# SOFT TOKENS, HARD TRUTHS

**Natasha Butt**[1,*]**, Ariel Kwiatkowski**[2]**, Ismail Labiad**[2]**, Julia Kempe**[2,3,†]**, Yann Ollivier**[2,†]

[1]University of Amsterdam    [2]Meta FAIR    [3]New York University

[*]Work done during an internship at Meta.    [†]Joint senior authors.

## ABSTRACT

The use of continuous instead of discrete tokens during the Chain-of-Thought (CoT) phase of reasoning LLMs has garnered attention recently, based on the intuition that a continuous mixture of discrete tokens could simulate a superposition of several reasoning paths simultaneously. Theoretical results have formally proven that continuous tokens have much greater expressivity and can solve specific problems more efficiently. However, practical use of continuous tokens has been limited by strong training difficulties: previous works either just use continuous tokens at inference time on a pre-trained discrete-token model, or must distill the continuous CoT from ground-truth discrete CoTs and face computational costs that limit the CoT to very few tokens.

This is the first work introducing a scalable method to learn continuous CoTs via reinforcement learning (RL), without distilling from reference discrete CoTs. We use "soft" tokens: mixtures of tokens together with noise on the input embedding to provide RL exploration. Computational overhead is minimal, enabling us to learn continuous CoTs with hundreds of tokens. On math reasoning benchmarks with Llama and Qwen models up to 8B, training with continuous CoTs match discrete-token CoTs for pass@1 and surpass them for pass@32, showing greater CoT diversity. In systematic comparisons, the best-performing scenario is to train with continuous CoT tokens then use discrete tokens for inference, meaning the "soft" models can be deployed in a standard way. Finally, we show continuous CoT RL training better preserves the predictions of the base model on out-of-domain tasks, thus providing a softer touch to the base model.

## 1    INTRODUCTION

Large Language Models (LLMs) have achieved impressive success across a wide range of reasoning tasks, particularly when enhanced with Chain-of-Thought (CoT) prompting, where models generate intermediate "thinking tokens" before producing final answers. While effective, standard CoT is constrained by the discreteness of language tokens: each intermediate step must be sampled sequentially, which can limit expressivity and hinder exploration of diverse reasoning paths. This contrasts sharply with human cognition, which often operates over abstract and fluid concepts rather than rigid linguistic symbols. Motivated by this gap, recent work has explored enabling LLMs to reason in continuous concept spaces, a direction often termed "continuous CoTs" (Hao et al., 2024) or "Soft Thinking" (Zhang et al., 2025).

From a theoretical perspective, continuous reasoning offers significant potential. *Reasoning by Superposition* (Zhu et al., 2025a) shows that continuous thought vectors can act as superposition states, encoding multiple search frontiers in parallel and enabling efficient breadth-first reasoning. This construction allows a shallow transformer to solve problems such as directed graph reachability far more efficiently than discrete CoT, which is forced into sequential exploration and risks being trapped in local solutions. Complementarily, *Soft Thinking* (Zhang et al., 2025) proposes replacing discrete ("hard") tokens with concept tokens—probability-weighted mixtures of embeddings—that retain full distributional information. This enables the model to implicitly follow multiple reasoning paths simultaneously, yielding empirical improvements in both accuracy and token efficiency.

Despite these promising claims, the practical benefits of continuous reasoning at inference time on top of discrete-token base models remain contested. In particular, Wu et al. (2025) critically

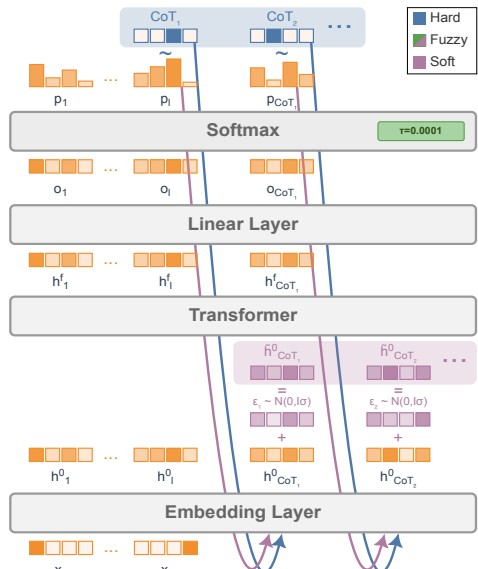

**Figure 1: Hard, fuzzy and soft generation during CoT phase**. In *hard* generation, at each time step, a discrete token $CoT_t$ is sampled from the probability vector $p_{t-1}$ and its embedding $h^0_{CoT_1}$ is passed to the transformer, generating a sequence of discrete CoT tokens: $CoT_1, ..., CoT_T$ over time. In *fuzzy* and *soft* generation, at each time step, noise, $\epsilon_t$, is injected into the probability weighted mixture embedding, $h^t_0 = p_{t-1}E$, where $E$ is the token embedding matrix. This noisy input embedding is passed to the transformer, generating a sequence of continuous noisy CoT embeddings: $\tilde{h}^0_{CoT_1}, ..., \tilde{h}^0_{CoT_T}$ over time. Additionally, for *fuzzy* generation, the temperature $\tau$ used in the CoT phase tends to 0, such that the non-noisy embeddings $h^0$ reduce to embeddings of discrete tokens. We find that the combination of soft/fuzzy training and hard inference typically performs best, often matching hard training at pass@1 and surpassing it at pass@32, indicating better preservation of diversity.

re-examine *Soft Thinking* and find that vanilla implementations often underperform their discrete counterparts. Their analysis suggests that LLMs, when given soft inputs, default to relying on the single highest-probability token—effectively reducing Soft Thinking to greedy decoding. Further, existing methods for soft thinking are limited to inference on models trained with discrete CoTs.

Training of continuous-token reasoning models has proven to be difficult, either due to computational constraints from full backpropagation through all steps of continuous reasoning (this limited the CoT to 6 steps in Hao et al. (2024)), or due to the necessity of strongly grounding the continuous reasoning into ground-truth discrete reasoning traces (Shen et al., 2025). This is why several of the works above limit themselves to applying continuous reasoning at inference time without training (Zhang et al., 2025; Wu et al., 2025).

In this work, we address these limitations by developing an approach to reinforce continuous CoTs with controlled noise, making them amenable to reinforcement learning (RL) training. We theoretically outline two types of continuous CoT learning with *soft* and *fuzzy* tokens (see Figure 1) and provide extensive empirical evidence with Llama-3.x and Qwen-2.5 models trained on a number of mathematical datasets (GSM8K, MATH, DeepScaleR) and evaluate on a variety of mathematical and out-of-domain benchmarks. Our contributions and findings are as follows:

- **A continuous-token post-training algorithm.** We propose the first continuous CoT fine-tuning algorithm that does not require ground-truth CoT annotations, directly tackling the challenge of learning continuous reasoning representations, at a negligible computational overhead compared to discrete CoTs.
- **Pass@1 parity.** We show that continuous CoT post-training is competitive with traditional discrete-token CoTs for pass@1 criteria, with a statistically similar performance on most model-dataset combinations.
- **Pass@32 gains.** Under sampling (pass@32), continuous CoT training outperforms discrete CoT on average, demonstrating greater CoT diversity.
- **Improved robustness.** Continuous CoT training does not degrade the base model's log-likelihood on HellaSwag, ARC and MMLU, whereas discrete CoT training does, on average. Further, continuous CoT training is robust to collapse observed with discrete training on Llama-8B-Instruct with respect to in-distribution and out-of-distribution performance.
- **Hard inference on soft models.** In our experiments, adding continuous CoT *at inference* on top of a hard-token-finetuned model does not bring benefits, contrasting with Zhang et al. (2025). On the opposite, the best performance is obtained when using discrete CoT at inference on top of a continuous CoT trained model. This means that a practitioner can deploy standard inference methods directly to reap the advantages from models trained with continuous tokens.

- **Entropy analysis.** We present a detailed analysis of the entropy profiles of models, showing how continuous CoT fine-tuning more closely preserves the entropy profiles of the base Llama models, compared to discrete CoT.

## 2 RELATED WORK

It is natural to question whether token space is the ideal medium for reasoning, particularly in tasks that demand higher levels of semantic abstraction. However, unrolling beyond token space during pretraining departs from the original training distribution and may require additional mechanisms to help a hard-token–trained LLM adapt to alternative, potentially more expressive, representations of chains of thought. Several prior works have confronted this challenge. (Goyal et al., 2024) proposes inserting dedicated placeholder tokens, "Pause Tokens", into rollouts to encourage more deliberate "thinking" in the internal layers. Coconut (Hao et al., 2024), in contrast, explicitly ventures into continuous-space reasoning by distilling ground-truth chains of thought into continuous tokens, but its benefits appear confined to a benchmark designed for this purpose. Furthermore, Coconut requires ground truth chains-of-thought and a gentle "hard-to-soft" distillation schedule, and comes with computational constraints that have limited it to 6 continuous CoT tokens.

A number of follow-up works have tried to tackle these challenges (see Zhu et al. (2025b) for a survey on latent-space reasoning). Given the frequent difficulty of making an LLM accept continuous tokens when they are not native to their training, it makes sense to distinguish *when* in the pipeline continuous tokens are invoked (at pretraining, at post-training or at inference). We list those works closest to ours, while necessarily being incomplete as the scope widens; we recommend the survey by Zhu et al. (2025b) for a taxonomy.

**Inference.** Zhang et al. (2025) introduce so-called *soft tokens* as the softmax layers of the output tokens, and propose to perform inference in soft-token space. Combined with a few other interventions (a hand-crafted stopping criterion, and only keeping the few top dimensions of embedding space) they seem to gain some performance on some benchmarks purely through inference-time interventions. A more measured view of the same is given in Wu et al. (2025), where the previous results are not confirmed, unless *noise* is introduced at test time into the soft token generation. Noise is important in our approach for a different reason, to provide exploration for reinforcement learning.

**Post-training.** A number of works aim to introduce continuous tokens during post-training as already mentioned for Coconut (Hao et al., 2024). Codi (Shen et al., 2025) distill a standard discrete CoT model into a continuous CoT model, by keeping both the emitted tokens and the internal activities of the continuous model close to that of the original model. Yue et al. (2025) introduce a curriculum by using a mixture of discrete and continuous tokens and slowly transitioning from discrete to continuous. Gozeten et al. (2025) use continuous tokens that are trained either via direct CoT supervision or via RL on a special noise model; scalability of their method is unclear as they test on a toy problem in which the CoT is a length-4 binary string, which allows for full enumeration of all 16 possible CoTs.

**Pretraining.** Several works propose changes to pretraining to incorporate some notion of "thinking tokens" or latent tokens, and attempt to replace the CoT by some internal activations inside, or added to, the transformer. "Filler tokens" approaches (Lanham et al., 2023; Goyal et al., 2024; Pfau et al., 2024; Ringel et al., 2025) introduce some bland tokens so that the model can use its continuous internal activations to reason while reading the bland tokens. Going further, CoCoMix (Tack et al., 2025) intersperses continuous tokens with hard tokens at pretraining, and uses a pretrained sparse autoencoder to couple the hard and soft tokens. "Looped transformers" (Saunshi et al., 2025) and "recurrent depth" (Geiping et al., 2025) deploy internal, continuous CoTs in the depth direction of the transformer, by repeating some internal blocks and making the CoT depth potentially infinite before every token.

**Theoretical arguments.** Zhu et al. (2025a) make a strong argument that continuous CoTs are more expressive than discrete CoTs, on a natural toy problem (reachability in graphs). They prove that a 2-layer transformer with continuous CoT can solve the problem in $O(n)$ vs $O(n^2)$ with discrete tokens. Continuous CoTs are provably able to use superpositions that explore several reasoning paths at the

same time. The experiments align with the theoretical predictions: the superposition is learned in practice. Gozeten et al. (2025) also includes, among other contributions, a theoretical argument showing that a different toy problem can be solved via a 1-layer transformer with continuous tokens, but not with discrete tokens.

**Diversity.** Our work crucially relies on introducing added noise in the continuous model: this produces the necessary variation upon which to apply RL methods such as GRPO. Without it, the continuous-token trajectory would be deterministic. Prior work has demonstrated the importance of diversity in reasoning trajectories; for example, Wang et al. (2023) leverages diverse reasoning paths sampled at inference time to improve robustness and accuracy via majority voting.

It is well established that adding noise can, by itself, improve performance of neural networks, for instance via dropout (Srivastava et al., 2014) or input perturbations as in denoising auto-encoders (Vincent et al., 2008). NEFTune and SymNoise (Jain et al., 2023; Yadav & Singh, 2023) apply this idea to the supervised fine-tuning phase of LLMs, by having them predict the unnoised sequence from a version of the sequence with noise added to the input embeddings. We use a similar noise structure.

Yet in our case, the benefits cannot come solely from the regularization effect of noise, since that would correspond to the noisy model at step 0 in our training curves, and the gains appear after training. Our approach explicitly optimizes the policy via RL to benefit from a diverse reasoning manifold during learning. The beneficial effect of added noise for boosting exploration and diversity in RL has been established: for instance, Fortunato et al. (2018) found benefits from adding noise directly to the weights of the value or policy networks in RL settings.

## 3 METHOD

**In a nutshell.** Similarly to Coconut or Soft Tokens, our method keeps the full probability distribution after the softmax step, instead of emitting a discrete token. This mixture of tokens is fed to the input of the transformer for the next step. Contrary to prior work, we then inject noise on the input embeddings. This noise produces the necessary exploration to apply RL fine-tuning. In contrast, Soft Tokens do not fine-tune, and Coconut relies on backpropagation through time through the whole continuous CoT (coming with strong computational limitations) plus a curriculum to ground the continuous CoTs into ground truth discrete CoTs. We now describe our method in detail.

**Notation for transformers.** We decompose a standard LLM architecture into the following blocks.

We denote by $V$ the vocabulary size. Thus, each token can be seen as a one-hot vector $x_t \in \mathbb{R}^V$, and the sequence of tokens $x_{<t}$ up to time $t$ can be seen as a matrix of size $t \times V$.

We denote by $E$ the token embedding matrix of an LLM, that takes a sequence of tokens $x_{<t}$ and returns a sequence of embeddings $h^0_{<t}$ by a linear, token-wise mapping

$$h^0_{<t} = x_{<t}E. \tag{1}$$

Denoting by $n_0$ the dimension of the input embedding of the transformer stack, $E$ is a matrix of size $V \times n_0$.

We denote by $\mathcal{T}$ the transformer stack, that turns the sequence of input embeddings $h^0_{<t}$ into a sequence of output embeddings

$$h^L_{<t} = \mathcal{T}(h^0_{<t}) \tag{2}$$

where $L$ is the depth of the transformer stack. $h^L_{<t}$ is a matrix of size $t \times n_L$ with $n_L$ the dimension of the output layer of the transformer stack.

Probabilities for next-token prediction are obtained as follows. The output encodings $h^L_{<t}$ are turned into logits by a decoding matrix $W_d$ of size $n_L \times V$ where $n_L$ is the output dimension of the transformer stack. The next-token probabilities are obtained by applying a softmax at temperature $\tau \geq 0$ to these logits:

$$p_{<t} = \mathrm{softmax}(h^L_{<t}W_d/\tau) \tag{3}$$

where $p_{<t}$ is a matrix of size $t \times V$, the softmax is applied for each $t$ independently, and softmax is extended by continuity for temperature $\tau = 0$.

**Standard (hard) tokens.** In standard, hard-token models, each token $x_t$ is a one-hot vector $x_t \in \mathbb{R}^V$. At inference time, to compute a next-token prediction $x_t$ given the sequence of previous tokens $x_{<t}$, one first computes the next-token probabilities $p_{<t}$ given $x_{<t}$ as above. Then the next token is sampled according to the last component $p_{t-1}$ of $p_{<t}$:

$$\Pr(x_t = 1_i) = p_{t-1,i} \tag{4}$$

where $1_i$ denotes the one-hot encoding of token $i$. This is applied inductively to get the sequence of next tokens.

**Soft thinking.** In soft thinking (Zhang et al., 2025; Wu et al., 2025), during the CoT phase, instead of sampling a next token $x_t$ according to the probabilities $p_t$, the probabilities are directly used to define a mixture of embeddings. The next input layer embedding is obtained as

$$h_t^0 = \sum_i \Pr(x_t = 1_i) e_i = p_{t-1} E \tag{5}$$

where $e_i$ is the embedding for token $i$. Then the transformer stack is applied normally to $h^0$. [1]

After the CoT phase is done, the model samples normal (hard) tokens.

This model is not amenable to direct RL training via Reinforce-like algorithms, because of the absence of noise or random choices: the whole CoT is a deterministic and differentiable function of the prompt. In principle, it could be optimized directly by backpropagating through all the timesteps of the CoT, similarly to Backpropagation Through Time (BPTT). But this leads to technical and memory challenges that we will not discuss here.

**Noisy soft thinking: soft tokens and fuzzy tokens.** Instead, we propose to make soft thinking trainable by RL, just by introducing noise into the soft thinking process. We simply add noise to the computation of $h_t^0$:

$$\tilde{h}_t^0 = p_{t-1} E + \sigma N(0, \text{Id}) \tag{6}$$

with some standard deviation $\sigma > 0$. Then, at the next timestep, the transformer stack is fed $\tilde{h}_t^0$.

We also experimented with adding noise at other places, such as on the logits (Appendix F.2).

We call this model *soft tokens*; we use the term *fuzzy tokens* when the temperature $\tau$ used during the chain of thought tends to 0, because in that case, the non-noisy embeddings $h^0$ reduce to embeddings of true discrete tokens, so $\tilde{h}^0$ are normal discrete tokens embeddings up to noise $\sigma$.

**Reinforcement learning on soft tokens.** Introducing exploration noise on the soft CoT tokens makes it possible to optimize the model via reinforcement learning. (For traditional discrete CoT tokens, exploration comes from the random sampling of a token from the softmax probabilities.)

We describe here the derivation of Reinforce for noisy soft thinking. More advanced Reinforce-like methods, such as RLOO, GRPO, PPO etc., are derived from Reinforce in the standard way.

Given a prompt, we sample a soft CoT, then sample a final answer $a$ given the CoT. Let $R(a)$ denote the reward obtained for an answer $a$. The objective is to maximize the expected reward.

The CoT sampling is fully defined by the sampling of the noisy soft tokens $\tilde{h}^0$. Therefore, the objective is to maximize the expectation

$$\mathbb{E}_{(\tilde{h},a) \sim \pi}[R(a)] \tag{7}$$

for a given prompt, where $\pi$ is the current model.

By the standard Reinforce theorem Sutton & Barto (1998), this is equivalent to minimizing the loss

$$\mathbb{E}_{(\tilde{h},a) \sim \pi^{\text{sg}}} \left[ -R(a) \left( \log \pi(\tilde{h}^0) + \log \pi(a|\tilde{h}^0) \right) \right] \tag{8}$$

---

[1]The model used in Coconut (Hao et al., 2024) is slightly different in that it directly feeds the output embedding as next-step input embeddings, namely, $h_t^0 = h_{t-1}^L$, assuming dimensions are the same. This bypasses the expansion from hidden dimension to vocabulary size and back, as well as the softmax and temperature.

The term $\log \pi(a|\tilde{h}^0)$ just represents fine-tuning the answer given the CoT, and can be computed in a standard way, since sampling of $a$ is done in a standard way.

The term $\log \pi(\tilde{h}^0)$ can be decomposed as a sum over timesteps,

$$\log \pi(\tilde{h}^0) = \sum_t \log \pi(\tilde{h}^0_t|\tilde{h}^0_{<t}) \tag{9}$$

and each of those terms can be computed easily: indeed, knowing the soft tokens $\tilde{h}^0_{<t}$, we can compute the non-noisy next-token input embedding $h^0_t$. Since the noise is Gaussian, we just have:

$$\log \pi(\tilde{h}^0_t|\tilde{h}^0_{<t}) = -\tfrac{1}{2\sigma^2} \left\| \tilde{h}^0_t - h^0_t \right\|^2 + \text{cst} \tag{10}$$

and we note that $h^0_t$ is a differentiable function of the previous soft tokens $\tilde{h}^0_{<t}$, depending on the parameters of the model.

This makes it possible to apply the family of Reinforce-like algorithms to noisy soft tokens.

Computational overhead is minimal: storing the probability vector $p_t$ at each step (vector of size $V$), and injecting noise on the first layer.

## 4 EXPERIMENTS

**Models tested.**   We train three variations of CoT models described in Section 3:

- Hard tokens: Categorical sampling of ordinary hard CoT tokens with temperature $\tau = 1.0$.
- Soft tokens: Instead of sampling hard tokens, we use the full probability mixture at temperature $\tau = 0.5$ to compute embeddings, and add Gaussian noise to the embeddings.
- Fuzzy tokens: Like soft tokens, but at temperature $\tau = 0.0001$, which brings them very close to hard tokens embeddings, to which we add Gaussian noise.

For the scale of the Gaussian noise, we set this equal to 0.33 times the root-mean-square norm of the token embeddings, so that the noise is comparable but a bit smaller than the embeddings. In practice we observe our algorithm is robust to ratios less than or equal to 1.0 (Appendix F.1). In further ablations (Appendix F.4), we observe that our algorithm is also robust to temperature values $\tau \in [0.0001, 0.1]$.

**Inference settings.**   At test time, we decouple the inference method from the training method: for each trained model (hard, soft, fuzzy), we evaluate six inference settings. We vary the decoding of the CoT as follows, but with answers always greedily decoded at temperature 0:

- Hard Greedy: discrete tokens, CoT temperature $\tau = 0.0$ at test time
- Hard Sample: discrete tokens, CoT temperature $\tau = 1.0$ at test time
- Soft Greedy: Gaussian scale $\sigma = 0.0$, CoT temperature $\tau = 0.5$ at test time
- Soft Sample: Gaussian scale $\sigma = 0.33 * \text{root-mean-square norm}$, CoT temperature $\tau = 0.5$ at test time
- Fuzzy Greedy: Gaussian scale $\sigma = 0.0$, CoT temperature $\tau = 0.0001$ at test time
- Fuzzy Sample: Gaussian scale $\sigma = 0.33 * \text{root-mean-square norm}$, CoT temperature $\tau = 0.0001$ at test time

For instance, "soft" training with "hard greedy" testing amounts to training with a mixture at $\tau = 0.5$ with Gaussian noise, then applying the model with hard tokens at test time.

The "sample" settings are the same as the variants used during training for the CoT, while the "greedy" setting more aggressively target the mode of the distribution at each step of the CoT.

**Reinforce with group baseline.**   We fine-tune the models with RLOO, namely, Reinforce using a per-prompt leave-one-out (LOO) group baseline (Kool et al., 2019): for each sample and reward, we subtract the average reward obtained on the other samples for the same prompt. We include the RLOO loss in Appendix A for completeness.

At each update we draw a mini-batch of $B = 2$ distinct prompts $\{x_b\}_{b=1}^B$. For each prompt we sample $G = 32$ sequences $y_{b,g}$ that contain a chain-of-thought (CoT) followed by a final answer. The prompt instructs the model to end the CoT with "The final answer is: " (see Appendix B).

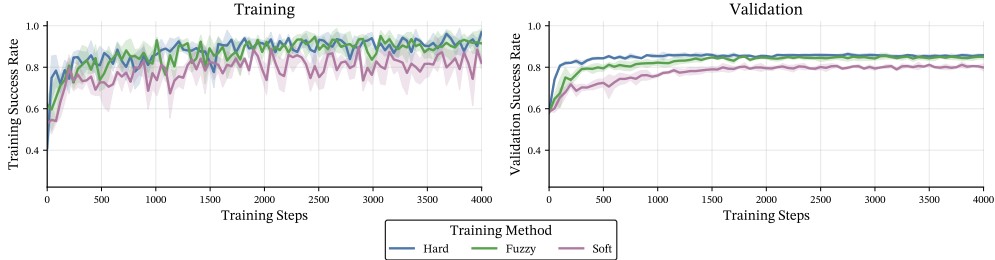

**Figure 2: Llama 3b Instruct trained on GSM8K** (a) Training performance across steps; one step = two prompts × 32 samples each. (b) Greedy validation performance used for model selection. For the remaining trained models, see Appendix G.1.

Rewards are computed only on the final answer using the *Math Verify* package (Kydlíček, 2025) against the ground-truth label:

$$
r_{b,g} = \begin{cases} 100, & \text{if } \mathrm{Verify}(a_{b,g}) = 1, \\ 10, & \text{if } \mathrm{Verify}(a_{b,g}) = 0 \text{ and } \mathrm{ExtractBoxed}(a_{b,g}) = 1, \\ 0, & \text{otherwise.} \end{cases} \tag{11}
$$

**Datasets and base models.** We train Llama 3.2 3b Instruct, Llama 3.1 8b Instruct (Dubey et al., 2024) and Qwen 2.5 3b Instruct (Yang et al., 2024) on math reasoning datasets including GSM8K (Cobbe et al., 2021), MATH (Hendrycks et al. (2021b) and DeepScaleR (Luo et al., 2025).

For each model trained on a dataset, we evaluate test performance on three math reasoning test datasets: GSM8K, MATH and OlympiadBench (He et al., 2024). For OlympiadBench, following prior work, we use the 675 subset of math questions which have final answers and do not contain images or figures. Similarly, for MATH, we evaluate on the MATH-500 (HuggingFaceH4, 2025) subset of the MATH test set. To assess out-of-distribution generalization, we also test the resulting models on standard benchmarks: HellaSwag (Zellers et al., 2019), MMLU (Hendrycks et al., 2021a), and ARC/AI2 Reasoning Challenge (Clark et al., 2018).

On each training dataset, we train for 4k steps and monitor greedy validation performance; the final model used for testing is the best performing under our greedy validation performance. (For hard CoTs, greedy performance refers to greedily decoding CoTs and answers. For soft and fuzzy CoTs, greedy performance refers to decoding CoTs with no Gaussian noise and greedily decoding answers.) During training, we sample a maximum of 128 CoT tokens for GSM8K and 512 CoT tokens for MATH and DeepScaler respectively, under our early stopping criterion (see Appendix C); on all datasets we sample 32 answer tokens. For evaluation, in all cases, we sample a maximum of 512 CoT tokens under our early stopping criterion followed by 32 answer tokens.

Each setup was run with 3 independent random seeds; the tables report the resulting mean and standard deviation. Training and validation success rates may be found in Figure 2. For details on hyper-parameters, see Appendix D.

**Results.** Across datasets and models, *the three training schemes are broadly comparable*, generally achieving similar greedy pass@1 performance as shown in Table 1. This demonstrates that fuzzy and soft training are effective. A notable exception is Llama-3B-Instruct trained on MATH, where hard CoT outperforms soft CoT by approximately 5% in pass@1. On the other hand, *soft and fuzzy training have a clear overall advantage for pass@32* over hard training (this signal is clearest on Llama, as shown in Figure 3).

We observe a gap between the *hard greedy* and *hard sample* pass@1 for the base models and models trained with fuzzy/soft CoTs, whereas the gap for models trained with hard CoTs is very small (see Figure 3 and Table 1). We note that, for the base and fuzzy/soft trained models, $\tau = 1$ is evidently not optimal for pass@1 (greedy $\tau = 0$ is always better). At low values of $k$, the optimal $\tau$ may be some interpolation between 0 and 1. Further, for pass@32, we also observe a closing of the gap between the trained models and base models. This loss of diversity is well reported for Reinforce style algorithms where the reward is based on answer correctness alone (Song et al., 2025).

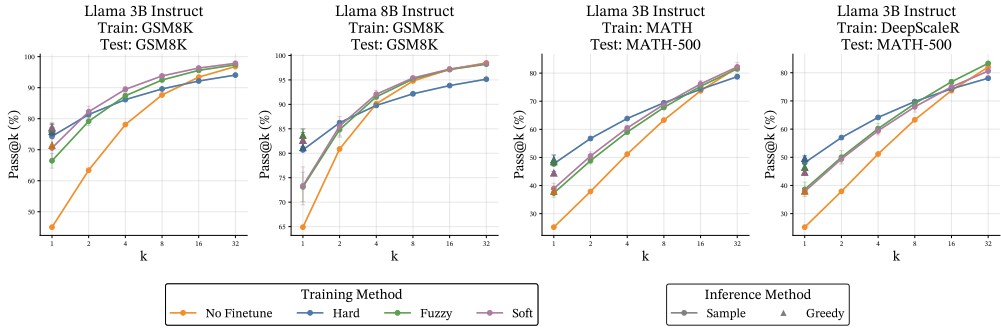

**Figure 3: Hard inference pass@k for Llama models** (for soft/fuzzy inference and Qwen see Appendix G.2). *We observe soft/fuzzy training improves pass@32, pointing to preserved diversity. Greedy Pass@1 (the triangles) for all training methods are clustered together.*

For all training methods (hard, soft or fuzzy), *hard inference generally performs best*, both for pass@1 and pass@32 (see Appendix E). In particular, we do *not confirm previously reported benefits of soft inference on hard (normal) training* (Zhang et al., 2025).

| Model | Training | GSM8K Greedy pass@1 | GSM8K Sample pass@1 | GSM8K Sample pass@32 | MATH-500 Greedy pass@1 | MATH-500 Sample pass@1 | MATH-500 Sample pass@32 | OlympiadBench Greedy pass@1 | OlympiadBench Sample pass@1 | OlympiadBench Sample pass@32 |
|---|---|---|---|---|---|---|---|---|---|---|
| | no finetune | 71.4±0.0 | 45.0±0.0 | 96.8±0.0 | 38.0±0.0 | 25.2±0.0 | 82.0±0.0 | 17.9±0.0 | 12.0±0.0 | 52.3±0.0 |
| llama 3b instruct | gsm8k hard | *75.9±1.3* | 74.3±0.8 | 94.1±0.3 | 34.6±0.9 | 31.3±0.4 | 72.4±1.1 | 10.6±0.9 | 9.4±0.5 | 39.3±1.3 |
| | gsm8k fuzzy | *76.7±1.8* | 66.4±2.4 | **97.4±0.3** | *42.6±3.2* | 32.4±1.7 | 83.2±1.1 | 17.4±0.9 | 13.0±1.1 | 55.2±1.2 |
| | gsm8k soft | *77.2±0.9* | 70.6±3.4 | **97.9±0.3** | *43.5±1.4* | 37.2±1.0 | **84.8±0.4** | *18.9±0.2* | 15.5±0.7 | **58.6±1.6** |
| | math hard | *80.0±0.5* | 79.7±0.8 | 96.7±0.1 | *49.1±1.7* | *47.8±0.9* | 78.7±0.9 | *22.7±1.1* | 22.2±0.4 | 49.3±1.3 |
| | math fuzzy | *79.6±1.4* | 68.5±2.1 | **97.6±0.3** | *48.5±0.3* | 37.3±1.5 | **81.5±0.7** | *21.6±0.6* | 15.9±0.8 | 52.8±1.6 |
| | math soft | 76.8±1.0 | 70.7±2.4 | **97.8±0.3** | 44.5±0.6 | 38.9±1.9 | **82.2±1.6** | 19.2±1.1 | 16.6±0.4 | **55.8±1.2** |
| | deepscaler hard | *79.7±1.3* | *79.6±0.2* | 96.6±0.3 | *49.6±0.9* | 48.1±0.4 | 78.1±0.7 | *23.2±1.4* | *23.8±0.6* | **50.8±1.5** |
| | deepscaler fuzzy | *78.8±0.8* | 69.8±4.7 | **97.7±0.5** | 46.5±2.2 | 38.6±2.6 | **83.3±0.3** | 19.5±1.0 | 16.6±1.4 | **54.4±2.1** |
| | deepscaler soft | *77.9±1.8* | 71.0±1.5 | **98.0±0.1** | 44.7±1.2 | 37.9±1.3 | 80.6±2.2 | 21.0±0.6 | 16.4±0.7 | 53.3±1.7 |
| | no finetune | 82.6±0.0 | 64.9±0.0 | 98.5±0.0 | 44.4±0.0 | 31.1±0.0 | 79.8±0.0 | 19.6±0.0 | 11.7±0.0 | 52.6±0.0 |
| llama 8b instruct | gsm8k hard | 81.2±0.2 | 80.7±0.3 | 95.1±0.2 | 20.2±0.8 | 19.8±1.4 | 45.4±3.2 | 3.8±0.5 | 3.4±0.5 | 16.4±2.4 |
| | gsm8k fuzzy | *83.7±1.3* | 73.1±3.0 | **98.2±0.2** | *44.6±2.1* | 33.8±2.7 | 83.1±0.9 | *18.0±1.4* | 12.5±1.3 | **56.0±2.6** |
| | gsm8k soft | *82.6±1.6* | 73.3±3.9 | **98.3±0.2** | *44.7±2.3* | 34.8±2.2 | 83.9±1.1 | *17.9±1.0* | 13.1±0.7 | **56.8±1.2** |
| qwen 3b instruct | no finetune | 8.9²±0.0 | 17.2±0.0 | 95.1±0.0 | 29.0±0.0 | 25.5±0.0 | 81.0±0.0 | 16.9±0.0 | 14.3±0.0 | 50.2±0.0 |
| | math hard | *84.0±0.8* | 83.0±0.2 | 97.2±0.3 | *59.0±1.7* | 57.1±0.1 | **83.6±1.0** | *29.4±0.3* | 27.8±0.5 | **61.1±0.6** |
| | math fuzzy | *84.4±0.8* | 81.6±0.9 | **98.1±0.2** | *58.1±0.9* | 55.5±1.2 | **84.4±0.2** | 27.2±0.5 | 24.4±0.7 | **60.7±0.5** |
| | math soft | *82.9±0.9* | 78.7±2.6 | **97.6±0.5** | 54.7±0.3 | 52.2±1.1 | **84.4±0.7** | 24.3±1.8 | 22.0±0.5 | 58.5±1.0 |

**Table 1: Results of hard inference on GSM8K, MATH-500 and OlympiadBench test sets**. In **blue** the best pass@1 performance and in **green** the best pass@32 for each (base model, training set) pair. *We observe broadly comparable performance at pass@1 and improved soft/fuzzy training pass@32.* For comparisons of hard, fuzzy and soft inference, see Tables 3, 4, 5 in Appendix E.

One setup stands out: *when training Llama-8B-Instruct on GSM8K and testing on MATH-500, only fuzzy and soft-trained models achieve good scores*, while classical hard token fine-tuning is ineffective. Namely, gsm8k-hard training sharply underperforms on out-of-distribution MATH (hard-greedy at 20.2% and pass@32 at 45.4%), whereas gsm8k-fuzzy and gsm8k-soft trainings recover to 44.6–44.7% greedy and 83.1–83.9% pass@32 while maintaining in-distribution performance on GSM8K. Llama-8B-Instruct has a good performance from the start on GSM8K (presumably because it was exposed to this dataset), but this does not translate to good performance on MATH. Further hard fine-tuning makes things worse, but further soft/fuzzy fine-tuning on GSM8K brings improvement on MATH. *Fuzzy/soft training appear to bring more generalization on Llama-8B-Instruct.*

**Out-of-domain robustness.** One risk of LLM fine-tuning on a dataset is degrading the general performance of the model on other datasets. To assess this, we test the trained models on three

standard benchmarks in Table 2. We report both the success rate (with hard greedy sampling) and the negative log-likelihood per token (NLL) of the correct answer.

| Model | Training | Hellaswag Accuracy | Hellaswag NLL Correct | ARC Accuracy | ARC NLL Correct | MMLU Accuracy | MMLU NLL Correct |
|---|---|---|---|---|---|---|---|
| | no finetune | 66.46±0.00 | 2.53±0.00 | 72.79±0.00 | 2.86±0.00 | 60.85±0.00 | 1.61±0.00 |
| llama 3b instruct | math hard | 66.20±0.10 | 2.57±0.00 | *73.45±0.43* | 3.19±0.03 | 61.11±0.11 | 1.65±0.01 |
| | math fuzzy | *67.00±0.12* | **2.53±0.00** | *73.25±0.32* | **2.88±0.03** | 61.16±0.04 | **1.60±0.02** |
| | math soft | *67.12±0.06* | **2.53±0.01** | 72.76±0.18 | **2.92±0.02** | *61.20±0.14* | **1.59±0.02** |
| | deepscaler hard | 66.18±0.26 | 2.58±0.01 | *73.99±0.57* | 3.18±0.03 | 60.66±0.37 | 1.66±0.01 |
| | deepscaler fuzzy | *66.88±0.06* | 2.55±0.01 | *73.05±0.62* | 2.94±0.03 | *61.20±0.20* | **1.60±0.02** |
| | deepscaler soft | *66.48±0.46* | **2.56±0.04** | 72.85±0.39 | **2.93±0.08** | *60.34±1.20* | 1.68±0.12 |
| | gsm8k hard | 66.44±0.18 | 2.57±0.00 | *73.53±0.25* | 3.16±0.01 | *61.48±0.20* | 1.65±0.01 |
| | gsm8k fuzzy | *66.97±0.21* | **2.53±0.01** | 73.10±0.15 | **2.89±0.01** | *61.30±0.21* | **1.60±0.02** |
| | gsm8k soft | *67.13±0.19* | **2.53±0.01** | 72.99±0.36 | **2.89±0.04** | *61.36±0.30* | **1.59±0.01** |
| | no finetune | 74.24±0.00 | 2.34±0.00 | 81.20±0.00 | 2.82±0.00 | 69.00±0.00 | 1.40±0.00 |
| llama 8b instruct | gsm8k hard | *74.44±0.10* | 2.37±0.00 | *81.49±0.15* | 3.13±0.03 | *68.90±0.04* | 1.41±0.00 |
| | gsm8k fuzzy | *74.41±0.03* | **2.35±0.00** | *81.63±0.39* | **2.81±0.01** | *68.90±0.11* | **1.40±0.00** |
| | gsm8k soft | *74.32±0.08* | **2.35±0.00** | *81.46±0.14* | **2.82±0.00** | 68.78±0.14 | **1.40±0.01** |
| | no finetune | 74.94±0.00 | 2.29±0.00 | 83.00±0.00 | 4.67±0.00 | 68.08±0.00 | 1.19±0.00 |
| qwen 3b instruct | math hard | 75.01±0.10 | 2.30±0.00 | *82.86±0.40* | 5.62±0.12 | 67.86±0.06 | 1.51±0.02 |
| | math fuzzy | 75.07±0.02 | **2.29±0.00** | *83.00±0.28* | **4.74±0.11** | *68.16±0.07* | **1.21±0.02** |
| | math soft | *75.25±0.12* | 2.30±0.01 | *83.12±0.15* | 5.49±0.64 | 68.01±0.06 | 1.33±0.06 |

**Table 2: Out-of-domain results**. *Outside of mathematical reasoning domains, fuzzy and soft training on average result in lower negative log-likelihood of the correct answer compared to hard training, indicating a softer touch on the base model's capabilities.*

The results show comparable success rate for the three training methods (hard, fuzzy, soft). However, the NLL of the correct answer is visibly better for fuzzy and soft than for hard, especially on ARC but also with Qwen on MMLU: *hard training degrades the base model NLL on out-of-domain datasets, while fuzzy and soft training preserve it.*

**Entropy behavior.** Next, we report the entropy of the distribution of next-token predictions during the CoT, as a function of the index within the CoT.

The base Llama models exhibit a very different entropy profile whether greedy or temperature sampling is used. This shows a difference in next-token prediction behavior on prefixes sampled from temperature $T = 0$ or $T = 1$: with the latter, entropy blows up as the CoT progresses with hard sampling, indicating very high uncertainty as the CoT goes on. Interestingly, *soft or fuzzy sampling on the base model does not show any entropy blowup*. The entropy blowup is also not present on Qwen, as seen in Appendix G.4 Figure 17. Analyzing base models is not our main topic, but we still report how this entropy profile changes after different types of training.

As observed in Figure 4 (and Figures 18, 19, 20 in Appendix G.4), *soft or fuzzy training keep roughly the same entropy profile as the base model*, whether inference is greedy or sampled. On the other hand, hard training changes the hard sampling entropy profile to resemble greedy sampling on the base model: entropy values are substantially lower. We pose that an explanation for this is that hard training makes the model overconfident, consistent with lower pass@32, the worse NLL values we observe on out-of-domain tasks (see Table 2) and the occasional performance collapse we see for hard training (see Table 1).

## 5  CONCLUSION

We have introduced the first reinforcement learning framework for training continuous Chains-of-Thought in LLMs with minimal computational overhead and without relying on ground-truth discrete CoTs. Across mathematical reasoning benchmarks, our approach performs on par with discrete token training for pass@1 success rate on most model–dataset combinations and improves pass@32 scores. Moreover, it seems to fine-tune the base model with a softer touch, better preserving the

---

[2]On Qwen no fine-tune, we observe much lower performance compared to what is reported in the Qwen report Yang et al. (2024). The discrepancy is due to our (zero shot) prompting and the resulting CoT generation format, and does not affect our RL trained models (see Appendix I for details).

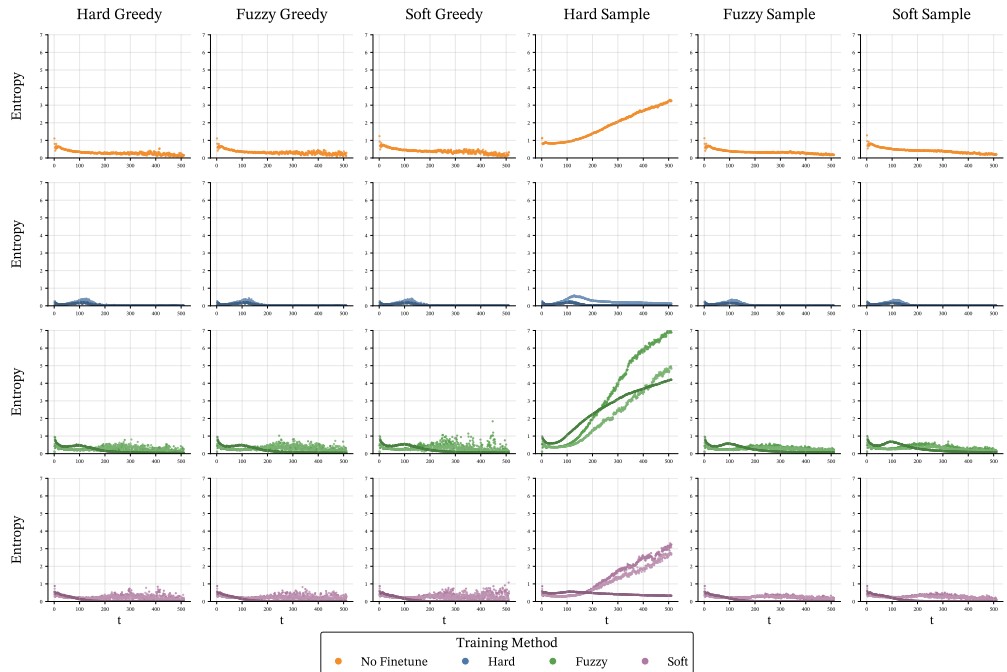

**Figure 4: Llama 3b Instruct CoT entropy on GSM8K test set**. *Fuzzy and soft training preserves entropy profile of base models; we observe a large change in hard sample profile with hard training.*

model's out-of-distribution behavior. This suggests distinct behavioral differences between soft and hard reasoning processes. These results provide evidence that continuous reasoning is not just a theoretical curiosity but a practical alternative for fine-tuning large models.

**Acknowledgments** We thank Shobhita Sundaram for the helpful discussions. JK acknowledges support by the Simons Foundation through the Collaborative Grant "The Physics of Learning and Neural Computation".

**Reproducibility Statement** To enable independent re-implementation, we provide in the paper and Appendix: (i) datasets and base models used for training and evaluation (ii) full hyperparameter details and training objective; (iii) prompt template and inference settings for evaluation; (iv) verifier used; (v) number of runs per result, with mean ± std; and (vi) compute details.

**LLM Usage** We used LLMs for drafting and polishing the writing. The research itself, and the bulk of the writing effort, were done by humans.

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

## A REINFORCE WITH LEAVE-ONE-OUT (RLOO) BASELINE

For completeness, we describe here the implementation of RLOO that we use, directly based on (Kool et al., 2019).

We fine-tune with Reinforce using a per-prompt leave-one-out (LOO) group baseline. At each update we draw a mini-batch of $B$ distinct prompts $\{x_b\}_{b=1}^{B}$. For each prompt we sample $G$ sequences $y_{b,g}$ that contain a chain-of-thought (CoT) followed by a final answer. Each such sequence provides a reward $r_{b,g}$ as described in Section 4.

For each prompt $x_b$, the LOO group baseline for sample $g$ averages the other $G-1$ rewards from the *same* prompt:

$$\bar{r}_b^{(-g)} \;=\; \frac{1}{G-1}\sum_{\substack{j=1\\j\neq g}}^{G} r_{b,j}, \qquad A_{b,g} \;=\; r_{b,g} - \bar{r}_b^{(-g)}. \tag{12}$$

Baselines and rewards are treated as constants w.r.t. $\theta$; i.e., stop-gradient.

Let $y_{b,g,1:T_{b,g}}$ denote all tokens in $y_{b,g}$ (CoT + answer). With policy $\pi_\theta$, the per-sequence log-probability is

$$\ell_{b,g} \;=\; \sum_{t=1}^{T_{b,g}} \log \pi_\theta\big(y_{b,g,t} \mid y_{b,g,<t}, x_b\big). \tag{13}$$

Our loss is the advantage-weighted negative log-likelihood averaged over batch and group:

$$\mathcal{L}(\theta) = -\frac{1}{BG}\sum_{b=1}^{B}\sum_{g=1}^{G} A_{b,g}^{\text{sg}}\,\ell_{b,g}, \tag{14}$$

$$\nabla_\theta \mathcal{L}(\theta) = -\frac{1}{BG}\sum_{b=1}^{B}\sum_{g=1}^{G} A_{b,g}\sum_{t=1}^{T_{b,g}} \nabla_\theta \log \pi_\theta\big(y_{b,g,t} \mid y_{b,g,<t}, x_b\big). \tag{15}$$

where $^{\text{sg}}$ denotes a stop-grad operator on the advantages.

## B TASK PROMPT

To guide the model's behavior during our experiments, we provided the following explicit instruction. This prompt ensures that the assistant follows a structured reasoning process before giving the final answer.

> **Task Prompt**
>
> ```
> A conversation between User and Assistant.  The user asks
> a question, and the Assistant solves it.  The assistant
> first shows the complete reasoning process step by step,
> then provides the final answer in \boxed{}.  The assistant
> must always follow the format:  'User:  [question] Assistant:
> [detailed reasoning] The final answer is:  \boxed{[answer]}.'
> User:  QUESTION Assistant:
> ```

## C STOPPING CRITERION AND PREFILLING

**Stopping criterion.**

- *Hard:* Monitor the generated text and stop as soon as it ends with `The final answer is:  `.

- *Soft/Fuzzy:* During continuous generation, form a greedy shadow sequence by taking the highest-probability hard token at each step and stop when this shadow ends with `The final answer is: `.

If neither condition is met, decoding continues until the maximum chain-of-thought length $L$.

**Prefilling.**
- If early stopping is reached, prefill `\boxed{`.
- If $L$ is reached, prefill `The final answer is: \boxed{`.

## D  HYPERPARAMETER SEARCH

All models were trained with the AdamW optimizer and a cosine learning-rate schedule using 20 warm up steps.

We tuned all hyperparameters using greedy validation performance. The same learning rate and scale factors are used for fuzzy and soft, thus we only sweep over values for fuzzy.

**Learning rate.**   For both the hard and fuzzy, we swept over $\{1e{-}5, 9e{-}6, \ldots, 2e{-}6, 1e{-}6\}$ for each combination of models. The same optimal rates were found for both hard and fuzzy:

- Llama 3B Instruct: $6e{-}6$
- Llama 8B Instruct: $3e{-}6$
- Qwen 3B Instruct: $8e{-}6$

**Scale factor (fuzzy only).**   We additionally swept scale factors $\{0.1, \ldots, 10\}$ on the root mean square embedding norm across various models and found $0.33$ to be best, though most values below 1 performed well. An ablation in Section F.1 supports the finding that our algorithm is robust to scale values below 1.

## E  RESULTS ON SOFT AND FUZZY INFERENCE

We evaluate base models and all RL trained models (hard, fuzzy and soft) under six inference settings: hard greedy, hard sample, fuzzy greedy, fuzzy sample, soft greedy and soft sample, described in Section 4. Here, we report test performance on GSM8K (Table 3), MATH-500 (Table 4) and OlympiadBench (Table 5). Contrary to previously reported benefits of soft inference on hard trained models (Zhang et al., 2025), we do not observe any improvement in soft inference. Interestingly, we do not observe any gains of soft inference on soft trained models either. In our experiments, hard inference on all models achieves the best performance.

## F  ABLATIONS

All ablations were run with two independent random seeds per experiment.

### F.1  NOISE SCALE

In Table 6, we report test performance on GSM8K test set for Llama 3b Instruct trained with the fuzzy RLOO, varying the scale factor, $\gamma$, applied to the root mean square embedding norm to compute the noise scale $\sigma$. Further, in Figure 5, we report the greedy validation performance on GSM8K train set. We observe that our algorithm appears robust to scale factors 1 and below; whereas for $\gamma = 3$, there is a collapse in learning as noise gets too large.

### F.2  NOISE PLACEMENT

In Table 7, we report test performance on GSM8K test set for Llama 3b Instruct trained with the fuzzy and soft RLOO, varying the placement of noise in our model. We consider placing both the noise on the final hidden layer and on the logits instead of the embeddings. We pose that the

| Model | Training | Inference Settings | | | | | | | | |
|---|---|---|---|---|---|---|---|---|---|---|
| | | Hard Greedy pass@1 | Hard Sample pass@1 | Hard Sample pass@32 | Fuzzy Greedy pass@1 | Fuzzy Sample pass@1 | Fuzzy Sample pass@32 | Soft Greedy pass@1 | Soft Sample pass@1 | Soft Sample pass@32 |
| | no finetune | 71.4±0.0 | 45.0±0.0 | 96.8±0.0 | 70.5±0.0 | 69.3±0.0 | 93.9±0.0 | 68.4±0.0 | 65.2±0.0 | 94.9±0.0 |
| llama 3b instruct | gsm8k hard | *75.9±1.3* | 74.3±0.8 | 94.1±0.3 | 75.5±0.6 | 74.7±0.5 | 92.3±0.4 | *75.7±0.5* | 74.2±0.4 | 92.6±0.1 |
| | gsm8k fuzzy | *76.7±1.8* | 66.4±2.4 | **97.4±0.3** | *76.4±2.1* | *75.2±1.8* | 92.0±1.1 | *75.1±1.8* | 73.5±1.7 | 93.5±0.8 |
| | gsm8k soft | *77.2±0.9* | 70.5±3.3 | **97.9±0.3** | *76.8±0.8* | *76.0±1.4* | 93.4±0.3 | 74.5±1.5 | 74.1±2.0 | 94.2±0.2 |
| | math hard | *80.0±0.5* | 79.7±0.8 | 96.7±0.1 | *80.1±0.5* | *78.9±0.7* | 95.3±0.5 | *79.5±0.8* | 78.6±0.7 | 95.5±0.3 |
| | math fuzzy | *79.6±1.4* | 68.5±2.1 | **97.6±0.3** | *78.8±0.9* | 78.1±0.8 | 95.2±0.1 | 77.9±0.9 | 75.9±1.1 | 95.7±0.3 |
| | math soft | 76.8±1.0 | 70.8±2.5 | **97.8±0.3** | 76.9±0.3 | 76.2±0.6 | 94.6±0.6 | 76.3±0.5 | 74.8±0.7 | 95.3±0.8 |
| | deepscaler hard | *79.7±1.3* | 79.5±0.3 | 96.6±0.3 | *79.6±1.1* | *78.9±0.5* | 94.7±0.6 | *79.8±1.0* | 78.5±0.5 | 94.8±0.2 |
| | deepscaler fuzzy | *78.8±0.8* | 69.6±4.7 | **97.7±0.5** | 78.3±0.9 | 77.2±1.1 | 94.8±0.4 | 76.5±0.7 | 75.4±1.3 | 95.3±0.6 |
| | deepscaler soft | *77.9±1.8* | 71.0±1.4 | **98.0±0.1** | 77.5±1.3 | 76.9±1.4 | 94.4±0.6 | *77.5±1.7* | 75.5±1.6 | 95.2±0.4 |
| llama 8b instruct | no finetune | 82.6±0.0 | 64.9±0.0 | 98.5±0.0 | 82.6±0.0 | 69.3±0.0 | 96.1±0.0 | 81.2±0.0 | 63.8±0.0 | 96.1±0.0 |
| | gsm8k hard | 81.2±0.2 | 80.6±0.4 | 95.1±0.2 | 80.0±1.3 | 69.0±0.7 | 94.5±0.3 | 78.3±2.2 | 68.3±0.7 | 94.3±0.1 |
| | gsm8k fuzzy | *83.7±1.3* | 73.2±3.0 | **98.2±0.2** | *83.4±0.7* | 73.5±1.1 | 95.5±0.3 | *82.3±0.8* | 69.8±1.7 | 95.5±0.7 |
| | gsm8k soft | *82.6±1.6* | 73.3±3.9 | **98.3±0.2** | *82.4±1.7* | 72.9±1.0 | 95.2±0.8 | *81.6±1.3* | 69.0±1.2 | 95.6±0.4 |
| qwen 3b instruct | no finetune | 8.9±0.0 | 17.2±0.0 | 95.1±0.0 | 10.5±0.0 | 11.5±0.0 | 58.5±0.0 | 8.6±0.0 | 13.8±0.0 | 71.9±0.0 |
| | math hard | *84.0±0.8* | 82.9±0.3 | 97.2±0.3 | *84.0±0.9* | 83.2±0.3 | 94.6±0.2 | *84.0±0.7* | 83.0±0.2 | 94.9±0.2 |
| | math fuzzy | *84.4±0.8* | 81.6±1.0 | **98.1±0.2** | *84.1±0.3* | *84.2±0.3* | 94.3±0.1 | *84.0±0.6* | *84.1±0.3* | 95.0±0.3 |
| | math soft | 82.9±0.9 | 78.7±2.6 | **97.6±0.5** | 82.5±1.3 | 82.1±1.3 | 94.8±0.4 | *82.9±1.1* | 81.9±1.5 | 95.4±0.4 |

**Table 3: Results on GSM8K test set**. In *blue* the best pass@1 performance and in **green** the best pass@32 for each (base model, training set) pair.

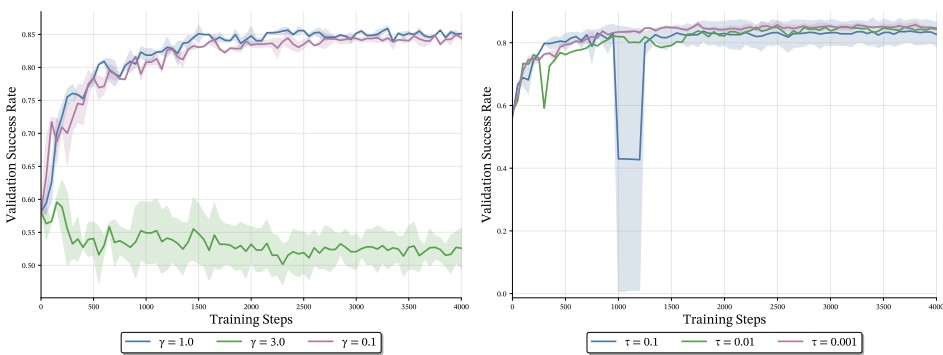

**Figure 5: Validation performance for: (a) noise scale ablation, (b) temperature ablation**, on Llama 3B Instruct trained with fuzzy models on GSM8K. *Fuzzy training appears robust to noise scale factors 0.1-1.0 and temperature values 0.1-0.0001.*

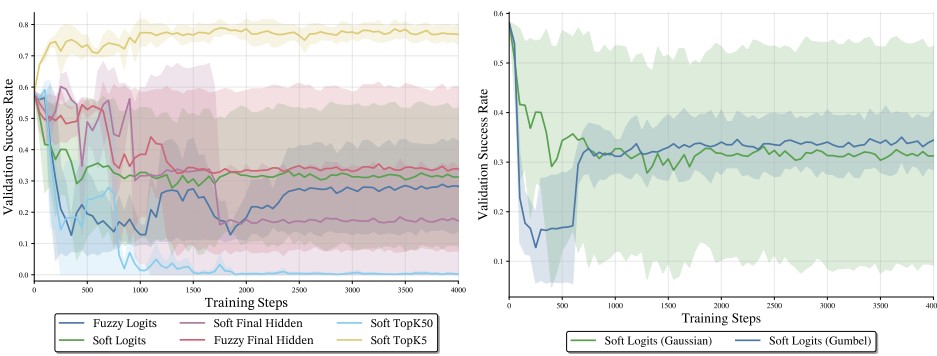

**Figure 6: Validation performance for (a) noise placement ablation (b) Gumbel noise ablation** on Llama 3B Instruct trained with fuzzy/soft models on GSM8K. *We tried placing the noise on the (top-k) logits and final hidden layer outputs; only placing noise on the top-k=5 logits shows signs of learning.*

| Model | Training | Inference Settings | | | | | | | | |
| | | Hard Greedy pass@1 | Hard Sample pass@1 | Hard Sample pass@32 | Fuzzy Greedy pass@1 | Fuzzy Sample pass@1 | Fuzzy Sample pass@32 | Soft Greedy pass@1 | Soft Sample pass@1 | Soft Sample pass@32 |
|---|---|---|---|---|---|---|---|---|---|---|
| | no finetune | 38.0±0.0 | 25.2±0.0 | 82.0±0.0 | 37.8±0.0 | 34.9±0.0 | 78.4±0.0 | 37.2±0.0 | 33.1±0.0 | 79.2±0.0 |
| llama 3b instruct | gsm8k hard | 34.6±0.2 | 31.2±0.4 | 72.4±1.1 | 34.6±0.0 | 32.7±0.5 | 71.6±0.9 | 33.2±1.6 | 32.1±0.8 | 70.1±0.2 |
| | gsm8k fuzzy | 42.6±3.2 | 32.6±1.6 | 83.2±1.1 | 41.6±2.1 | 41.2±2.0 | 79.5±1.7 | 40.3±1.5 | 38.7±1.7 | 79.5±2.0 |
| | gsm8k soft | 43.5±1.4 | 37.0±1.0 | 84.8±0.4 | 44.2±0.5 | 42.5±0.7 | 78.1±0.4 | 42.0±2.3 | 40.7±0.8 | 79.6±1.0 |
| | math hard | 49.1±1.7 | 47.9±0.9 | 78.7±0.9 | 49.7±1.0 | 47.0±1.0 | 76.5±0.2 | 48.1±1.7 | 46.8±1.0 | 76.3±1.0 |
| | math fuzzy | 48.5±0.3 | 37.5±1.6 | 81.5±0.7 | 47.4±0.7 | 46.9±0.9 | 79.7±0.6 | 45.6±1.6 | 44.8±0.6 | 79.9±0.1 |
| | math soft | 44.5±0.6 | 39.1±1.8 | 82.2±1.6 | 45.1±1.4 | 43.4±0.6 | 79.3±1.7 | 43.2±0.8 | 42.0±0.3 | 80.2±0.9 |
| | deepscaler hard | 49.6±0.9 | 48.3±0.6 | 78.1±0.7 | 49.2±0.7 | 47.9±0.4 | 77.7±0.5 | 50.0±0.6 | 47.4±0.2 | 77.7±1.1 |
| | deepscaler fuzzy | 46.5±2.2 | 38.5±2.6 | 83.3±0.3 | 46.0±1.7 | 45.3±1.2 | 79.8±0.3 | 45.8±0.9 | 43.5±1.4 | 78.9±0.5 |
| | deepscaler soft | 44.7±1.2 | 37.4±1.3 | 80.6±2.2 | 44.6±1.1 | 42.8±0.7 | 78.5±1.2 | 40.9±1.4 | 41.3±1.6 | 77.9±1.2 |
| llama 8b instruct | no finetune | 44.4±0.0 | 31.1±0.0 | 79.8±0.0 | 45.2±0.0 | 32.2±0.0 | 78.6±0.0 | 42.4±0.0 | 29.3±0.0 | 73.6±0.0 |
| | gsm8k hard | 20.2±0.8 | 19.8±1.2 | 45.4±3.2 | 20.8±1.0 | 9.7±0.6 | 32.3±0.2 | 20.0±1.8 | 9.7±0.6 | 32.2±0.4 |
| | gsm8k fuzzy | 44.6±2.1 | 33.7±2.5 | 83.1±0.9 | 45.1±1.4 | 33.9±0.9 | 75.9±2.1 | 43.7±0.8 | 31.1±1.2 | 75.7±2.1 |
| | gsm8k soft | 44.7±2.3 | 34.8±2.2 | 83.9±1.1 | 44.3±1.7 | 33.9±0.7 | 76.9±1.4 | 44.3±0.9 | 31.0±0.5 | 75.4±1.1 |
| qwen 3b instruct | no finetune | 29.0±0.0 | 25.5±0.0 | 81.0±0.0 | 27.6±0.0 | 28.1±0.0 | 69.0±0.0 | 27.0±0.0 | 29.6±0.0 | 73.6±0.0 |
| | math hard | 59.0±1.7 | 57.1±0.2 | 83.6±1.0 | 58.9±1.9 | 57.5±0.4 | 79.9±1.0 | 58.3±0.2 | 57.1±0.5 | 80.2±0.7 |
| | math fuzzy | 58.1±0.9 | 55.5±1.2 | 84.4±0.2 | 58.4±0.8 | 57.9±0.8 | 79.1±0.7 | 58.0±1.4 | 57.7±1.0 | 80.3±0.8 |
| | math soft | 54.7±0.3 | 52.2±1.1 | 84.4±0.7 | 54.9±0.5 | 54.3±0.1 | 79.1±0.5 | 54.7±1.2 | 54.6±0.6 | 80.3±1.5 |

**Table 4: Results on MATH-500**. In *blue* the best pass@1 performance and in **green** the best pass@32 for each (base model, training set) pair.

| Model | Training | Inference Settings | | | | | | | | |
| | | Hard Greedy pass@1 | Hard Sample pass@1 | Hard Sample pass@32 | Fuzzy Greedy pass@1 | Fuzzy Sample pass@1 | Fuzzy Sample pass@32 | Soft Greedy pass@1 | Soft Sample pass@1 | Soft Sample pass@32 |
|---|---|---|---|---|---|---|---|---|---|---|
| | no finetune | 17.9±0.0 | 12.0±0.0 | 52.3±0.0 | 18.4±0.0 | 15.4±0.0 | 53.2±0.0 | 14.4±0.0 | 14.7±0.0 | 52.4±0.0 |
| llama 3b instruct | gsm8k hard | 10.6±0.9 | 9.4±0.5 | 39.3±1.3 | 11.0±1.2 | 10.1±0.4 | 39.1±0.8 | 11.1±0.5 | 9.8±0.3 | 38.0±1.9 |
| | gsm8k fuzzy | 17.4±0.9 | 13.0±1.1 | 55.2±1.2 | 17.6±1.3 | 16.4±1.1 | 55.3±2.1 | 15.5±0.3 | 15.5±0.6 | 55.2±1.5 |
| | gsm8k soft | 18.9±0.2 | 15.5±0.7 | 58.6±1.6 | 19.3±1.2 | 18.2±0.4 | 57.5±0.8 | 17.4±0.4 | 16.9±0.2 | 57.5±0.7 |
| | math hard | 22.7±1.1 | 22.2±0.4 | 49.3±1.3 | 22.2±1.2 | 22.1±0.3 | 48.5±1.5 | 22.7±0.7 | 21.9±0.4 | 48.0±1.4 |
| | math fuzzy | 21.6±0.6 | 15.9±0.8 | 52.8±1.6 | 21.2±1.1 | 21.0±0.7 | 55.4±1.7 | 19.9±0.4 | 19.6±0.5 | 56.2±0.6 |
| | math soft | 19.2±1.1 | 16.6±0.4 | 55.8±1.2 | 20.2±0.5 | 19.5±0.4 | 53.5±2.1 | 19.9±0.7 | 18.2±0.3 | 53.9±1.1 |
| | deepscaler hard | 23.2±1.4 | 23.8±0.6 | 50.8±1.5 | 23.0±0.6 | 23.7±0.3 | 49.4±1.1 | 24.0±1.2 | 23.3±0.5 | 49.0±1.9 |
| | deepscaler fuzzy | 19.5±1.0 | 16.6±1.4 | 54.4±2.1 | 20.7±1.9 | 19.7±0.9 | 53.1±1.0 | 20.5±1.1 | 18.5±1.1 | 53.8±1.3 |
| | deepscaler soft | 21.0±0.6 | 16.4±0.7 | 53.3±1.7 | 18.9±1.3 | 19.5±0.1 | 53.2±1.5 | 18.4±0.3 | 18.0±0.4 | 51.5±1.6 |
| llama 8b instruct | no finetune | 19.6±0.0 | 11.7±0.0 | 52.6±0.0 | 18.7±0.0 | 12.8±0.0 | 46.4±0.0 | 15.7±0.0 | 11.5±0.0 | 48.3±0.0 |
| | gsm8k hard | 3.8±0.5 | 3.4±0.5 | 16.4±2.4 | 3.8±0.8 | 1.2±0.2 | 7.6±0.4 | 3.9±0.1 | 1.0±0.2 | 6.7±0.8 |
| | gsm8k fuzzy | 18.0±1.4 | 12.5±1.3 | 56.0±2.6 | 17.9±1.3 | 14.1±1.3 | 52.5±2.9 | 18.1±1.4 | 13.0±1.4 | 51.5±2.7 |
| | gsm8k soft | 17.9±1.0 | 13.1±0.7 | 56.8±1.2 | 18.7±1.3 | 13.8±0.4 | 53.0±1.5 | 17.1±0.6 | 13.0±0.5 | 51.1±0.9 |
| qwen 3b instruct | no finetune | 16.9±0.0 | 14.3±0.0 | 50.2±0.0 | 17.3±0.0 | 16.4±0.0 | 40.6±0.0 | 17.3±0.0 | 16.7±0.0 | 45.3±0.0 |
| | math hard | 29.4±0.3 | 27.8±0.5 | 61.1±0.6 | 29.6±0.8 | 28.6±0.4 | 57.2±0.4 | 29.1±0.7 | 28.3±0.6 | 55.7±0.5 |
| | math fuzzy | 27.2±0.5 | 24.4±0.7 | 60.7±0.5 | 27.4±1.2 | 27.1±0.5 | 54.0±0.1 | 27.2±0.7 | 26.6±0.7 | 55.4±0.1 |
| | math soft | 24.3±1.8 | 22.0±0.5 | 58.5±1.0 | 22.8±1.5 | 23.9±0.6 | 53.2±2.9 | 23.1±2.3 | 23.5±0.4 | 54.9±1.7 |

**Table 5: Results on OlympiadBench**. In *blue* the best pass@1 performance and in **green** the best pass@32 for each (base model, training set) pair.

dimensionality of the noise on the logits layer may be too high for learning given the signal to noise ratio and so consider a top-k variant where we only add noise to the top-k logits and set the remaining logits to negative infinity. In Figure 6, we report the greedy validation performance on GSM8K train set. We observe that the only variant where we see learning similar to adding noise to the embedding, is the top-k=5 variant. Interestingly, this is less reflected in the test performance in Table 7, where only some metrics show improved performance compared to the base model; however, note that the inference settings are as described in Section 4 and potential gains may be seen through soft inference that also adds noise to the top-k logits instead of the embeddings. Despite a collapse in validation performance, the top-k 50 test performance is on average higher than the base model suggesting some learning, however, on inspection the model checkpoints with the best validation performance which are used for evaluation were those after 50-100 steps.

| Noise Factor | Inference setting | | | | | | | | |
| --- | --- | --- | --- | --- | --- | --- | --- | --- | --- |
| | Hard Greedy pass@1 | Hard Sample pass@1 | Hard Sample pass@32 | Fuzzy Greedy pass@1 | Fuzzy Sample pass@1 | Fuzzy Sample pass@32 | Soft Greedy pass@1 | Soft Sample pass@1 | Soft Sample pass@32 |
| $\gamma$=0.1 | 75.5±0.8 | 68.9±3.2 | 97.5±0.1 | 74.5±0.9 | 74.5±1.1 | 92.9±0.3 | 73.5±1.2 | 72.9±1.5 | 93.9±0.2 |
| $\gamma$=0.33 | 76.7±1.8 | 66.4±2.4 | 97.4±0.3 | 76.4±2.1 | 75.2±1.8 | 92.0±1.1 | 75.1±1.8 | 73.5±1.7 | 93.5±0.8 |
| $\gamma$=1.0 | 78.1±0.2 | 71.7±0.1 | 97.7±0.2 | 77.7±0.0 | 77.0±0.1 | 93.3±0.5 | 77.5±0.5 | 75.8±0.3 | 94.4±0.6 |
| $\gamma$=3.0 | 65.4±1.9 | 37.8±4.4 | 95.9±0.6 | 65.1±1.7 | 63.1±2.1 | 91.1±1.3 | 60.5±2.9 | 57.3±3.0 | 93.1±0.6 |

**Table 6: Results of noise factor ablation study** on GSM8K test set of for Llama 3B Instruct trained with fuzzy Model on GSM8K train. *Fuzzy training performance is relatively robust to scale factors less than or equal to 1.0.*

| Training | Noise Loc | Temp | Top-k | Inference setting | | | | | | | | |
| --- | --- | --- | --- | --- | --- | --- | --- | --- | --- | --- | --- | --- |
| | | | | Hard Greedy pass@1 | Hard Sample pass@1 | Hard Sample pass@32 | Fuzzy Greedy pass@1 | Fuzzy Sample pass@1 | Fuzzy Sample pass@32 | Soft Greedy pass@1 | Soft Sample pass@1 | Soft Sample pass@32 |
| None | Tokens | 1.0 | 1 | 71.0±0.0 | 45.0±0.0 | 96.8±0.0 | 70.8±0.0 | 69.3±0.0 | 93.9±0.0 | 67.7±0.0 | 65.2±0.0 | 94.9±0.0 |
| Fuzzy | Embedding | 0.0001 | - | 76.7±1.8 | 66.4±2.4 | 97.4±0.3 | 76.4±2.1 | 75.2±1.8 | 92.0±1.1 | 75.1±1.8 | 73.5±1.7 | 93.5±0.8 |
| Soft | Embedding | 0.5 | - | 77.2±0.9 | 70.6±3.4 | 97.9±0.3 | 76.8±0.8 | 76.0±1.3 | 93.4±0.3 | 74.5±1.5 | 74.1±2.1 | 94.2±0.2 |
| Fuzzy | Final Hidden | 0.0001 | - | 66.2±4.1 | 46.6±12.4 | 96.1±1.0 | 66.0±3.5 | 64.6±4.1 | 89.2±0.9 | 62.7±4.6 | 60.1±5.8 | 90.6±0.6 |
| Soft | Final Hidden | 0.5 | - | 68.0±3.3 | 40.0±10.0 | 94.5±1.7 | 66.8±3.5 | 65.1±3.6 | 89.3±1.0 | 64.3±3.3 | 61.0±3.7 | 91.7±0.6 |
| Fuzzy | Logits | 0.0001 | - | 66.8±2.4 | 32.3±2.7 | 94.3±0.8 | 65.2±3.5 | 63.9±2.9 | 91.3±2.2 | 61.1±4.2 | 56.8±4.3 | 94.0±1.4 |
| Soft | Logits | 0.5 | - | 60.0±14.3 | 35.8±12.6 | 94.3±2.1 | 59.0±13.6 | 58.6±12.9 | 90.0±3.3 | 56.8±13.0 | 53.8±13.4 | 92.4±2.6 |
| Soft | Logits | 1.0 | 5 | 72.8±0.1 | 68.5±0.7 | 94.0±0.7 | 72.1±0.2 | 71.5±0.6 | 90.9±0.5 | 71.6±0.6 | 70.7±0.6 | 91.5±0.4 |
| Soft | Logits | 1.0 | 50 | 74.2±0.2 | 58.4±5.8 | 97.5±0.6 | 73.4±0.5 | 71.9±0.1 | 94.2±0.0 | 72.0±0.3 | 68.7±0.3 | 94.7±0.3 |

**Table 7: Results of noise placement ablation study** on GSM8K test set for Llama 3B Instruct trained with fuzzy/soft models on GSM8K train. *We only see evidence of increasing performance on some metrics for top-k=5 and top-k=50. Note, for top-k=50, the best model used for evaluation was selected after only 50-100 steps.*

### F.3 GUMBEL NOISE

Here, we extend the noise-placement variant in Appendix F.2 that places Gaussian noise on the logits. For connectivity with Wu et al. (2025), we instead place Gumbel noise on the logits and apply soft training. In Figure 6, we observe no evidence of learning for either variant that places noise on the logits, whether that noise is Gaussian or Gumbel.

| Training | Noise Loc | Temp | Noise Type | Inference setting | | | | | | | | |
| --- | --- | --- | --- | --- | --- | --- | --- | --- | --- | --- | --- | --- |
| | | | | Hard Greedy pass@1 | Hard Sample pass@1 | Hard Sample pass@32 | Fuzzy Greedy pass@1 | Fuzzy Sample pass@1 | Fuzzy Sample pass@32 | Soft Greedy pass@1 | Soft Sample pass@1 | Soft Sample pass@32 |
| None | Tokens | 1.0 | - | 71.0±0.0 | 45.0±0.0 | 96.8±0.0 | 70.8±0.0 | 69.3±0.0 | 93.9±0.0 | 67.7±0.0 | 65.2±0.0 | 94.9±0.0 |
| Soft | Embedding | 0.5 | Gaussian | 77.2±0.9 | 70.6±3.4 | 97.9±0.3 | 76.8±0.8 | 76.0±1.3 | 93.4±0.3 | 74.5±1.5 | 74.1±2.1 | 94.2±0.2 |
| Soft | Logits | 0.5 | Gaussian | 60.0±14.3 | 35.8±12.6 | 94.3±2.1 | 59.0±13.6 | 58.6±12.9 | 90.0±3.3 | 56.8±13.0 | 53.8±13.4 | 92.4±2.6 |
| Soft | Logits | 0.5 | Gumbel | 66.0±5.8 | 40.5±7.4 | 96.1±0.7 | 48.9±9.2 | 22.8±0.0 | 85.1±1.4 | 48.9±9.2 | 22.8±0.0 | 85.5±1.1 |

**Table 8: Results of gumbel noise ablation study** on GSM8K test set for Llama 3B Instruct trained with soft models on GSM8K train. *We see no evidence of increasing performance for either the Gaussian or Gumbel noise variants on logits.*

### F.4 TEMPERATURE

In Table 9, we report test performance on GSM8K test set for Llama 3b Instruct trained with the fuzzy RLOO, varying the temperature, $\tau$. Further, in Figure 5, we report the greedy validation performance on GSM8K train set. We observe that our algorithm appears robust to temperatures between 0.1 and 0.0001.

## G  SUPPLEMENTARY RESULTS

### G.1 TRAINING AND VALIDATION PERFORMANCES

In Figures 7, 8, 9 and 10, we report training and validation success rates for all RL trained models.

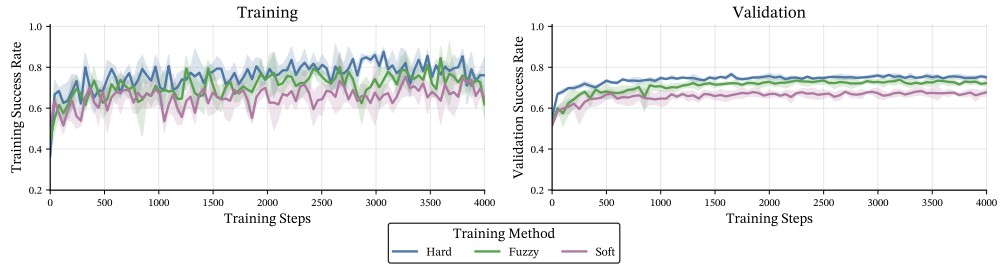

**Figure 7: Llama 3b Instruct trained on MATH** (a) Training performance across steps. (b) Greedy validation performance used for model selection.

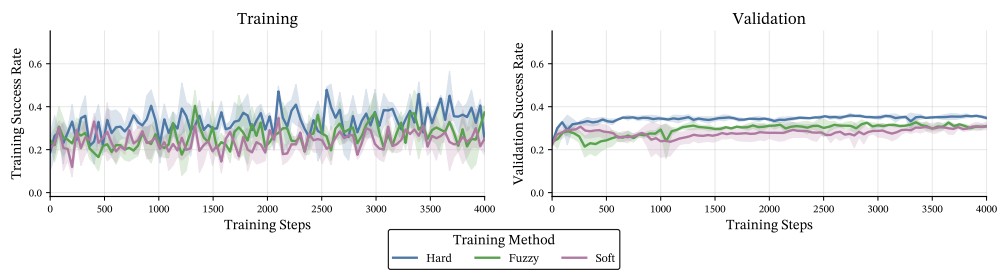

**Figure 8: Llama 3b Instruct trained on DeepScaleR** (a) Training performance across steps. (b) Greedy validation performance used for model selection.

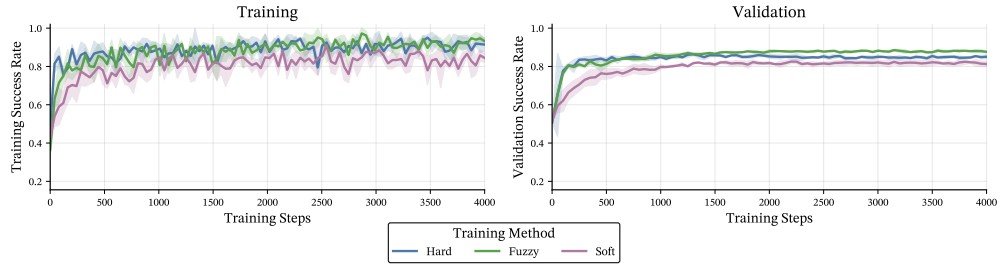

**Figure 9: Llama 8b Instruct trained on GSM8K** (a) Training performance across steps. (b) Greedy validation performance used for model selection.

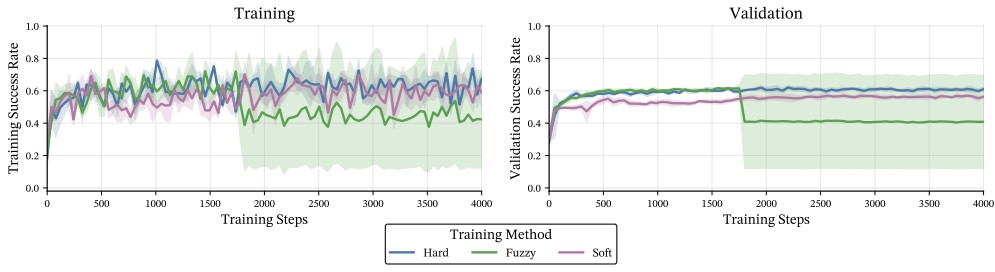

**Figure 10: Qwen 3b Instruct trained on MATH** (a) Training performance across steps. (b) Greedy validation performance used for model selection.

| Temperature | Inference setting | | | | | | | | |
|---|---|---|---|---|---|---|---|---|---|
| | Hard Greedy pass@1 | Hard Sample pass@1 | Hard Sample pass@32 | Fuzzy Greedy pass@1 | Fuzzy Sample pass@1 | Fuzzy Sample pass@32 | Soft Greedy pass@1 | Soft Sample pass@1 | Soft Sample pass@32 |
| $\tau$=0.0001 | 76.7±1.8 | 66.4±2.4 | 97.4±0.3 | 76.4±2.1 | 75.2±1.8 | 92.0±1.1 | 75.1±1.8 | 73.5±1.7 | 93.5±0.8 |
| $\tau$=0.001 | 77.4±0.3 | 67.8±1.5 | 97.8±0.2 | 77.3±0.1 | 76.2±0.4 | 92.2±0.2 | 76.5±1.2 | 74.6±0.1 | 93.9±0.6 |
| $\tau$=0.01 | 76.9±0.2 | 70.7±1.1 | 97.3±0.0 | 77.2±0.1 | 76.2±0.4 | 93.7±0.1 | 75.5±0.2 | 74.6±0.5 | 94.2±0.3 |
| $\tau$=0.1 | 76.3±2.0 | 63.1±7.5 | 97.4±0.2 | 75.1±2.3 | 74.5±2.5 | 94.1±2.5 | 74.3±3.3 | 72.4±2.7 | 94.4±2.4 |

**Table 9: Results of temperature ablation study** on GSM8K Test Set for Llama 3B Instruct Trained with Fuzzy Model on GSM8K Train. *Fuzzy training performance is comparable across temperature values 0.1-0.0001.*

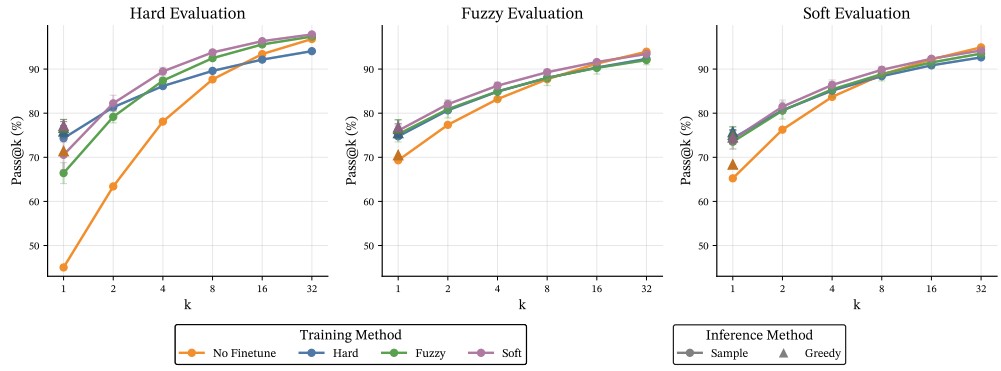

**Figure 11: Pass@k on GSM8K test set of Llama 3b Instruct trained on GSM8K train**

## G.2 PASS@K

In Figures 11, 12, 13, 14, 15, we report the pass@k on each model and training dataset combination across each training method (none, hard, fuzzy and soft) and evaluation metric (hard, fuzzy, soft).

## G.3 PASS@64

We extend our pass@k analysis for Llama 3b Instruct trained on GSM8K to pass@64 using one seed for each evaluation. In Figure 16, we see that for hard inference, at pass@64, soft and fuzzy training methods achieve similar performance to the no-finetune base model, whereas the hard training method continues to under-perform.

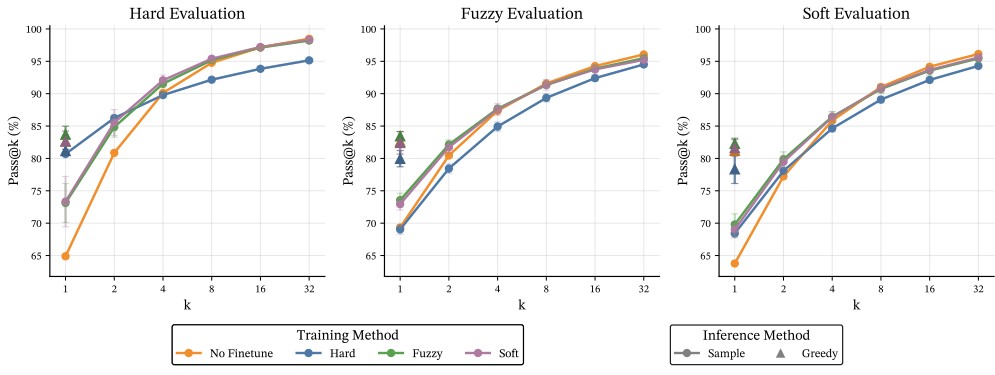

**Figure 12: Pass@k on GSM8K test set of Llama 8b Instruct trained on GSM8K train**

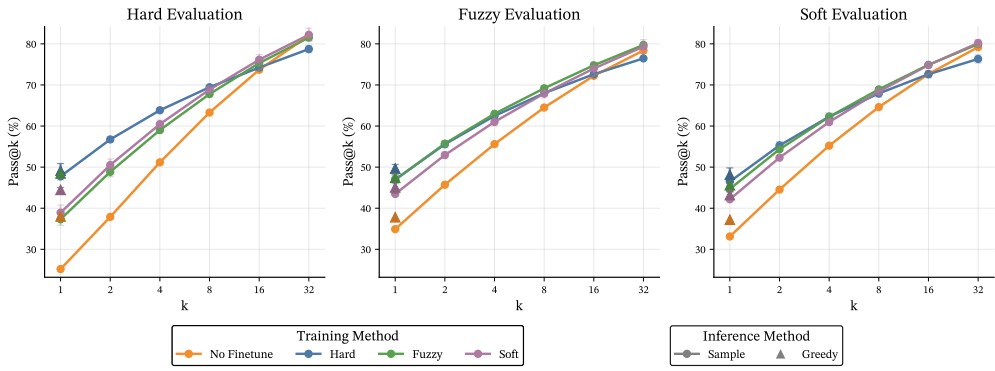

**Figure 13: Pass@k on MATH-500 of Llama 3b Instruct trained on MATH train**

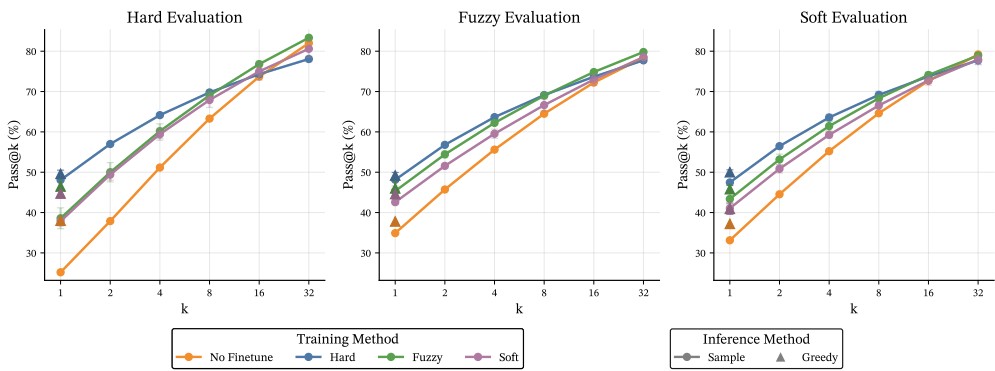

**Figure 14: Pass@k on MATH-500 of Llama 3b Instruct trained on DeepScaleR**

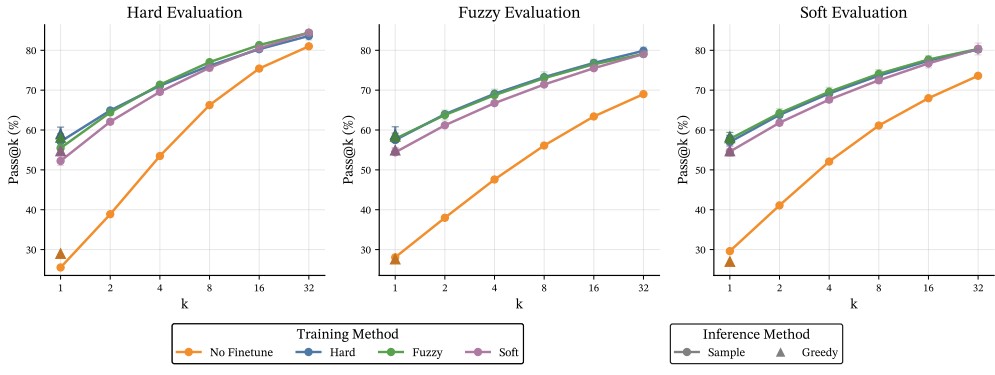

**Figure 15: Pass@k on MATH-500 of Qwen 3b Instruct trained on MATH train**

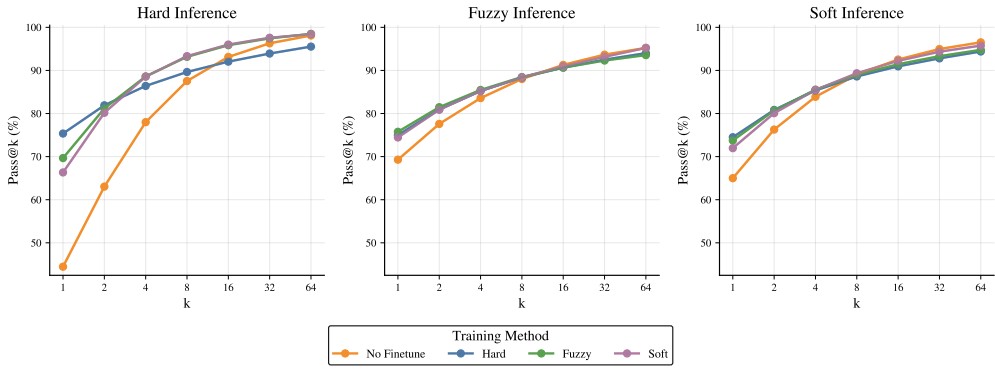

**Figure 16: Extended pass@k on GSM8K test set of Llama 3b Instruct trained on GSM8K train**

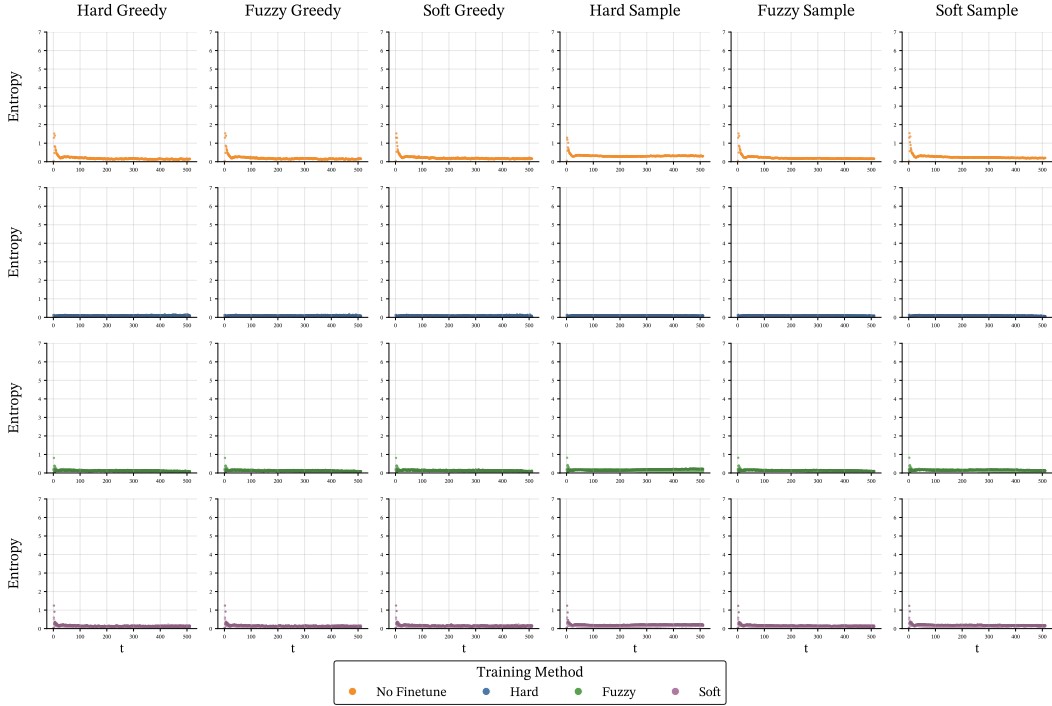

**Figure 17: CoT Entropy on MATH-500 of Qwen 3b trained on MATH train**

### G.4 ENTROPY ANALYSIS

In Figures 17, 18, 19, 20, we report the CoT entropy on each model and training dataset combination across each training method (none, hard, fuzzy and soft) and evaluation metric (hard greedy, hard sample, fuzzy greedy, hard greedy, soft greedy, soft sample). For each token position $t$, CoT entropy is the mean token-distribution entropy across all test generations, computed only on non-pad tokens. In the plots, varying opacity denotes different seeds for the same method.

## H COMPUTATION DETAILS

Every RLOO run (hard, fuzzy, and soft) was executed on a dedicated node with 8× NVIDIA H100 GPUs (80 GB VRAM each) or 8× NVIDIA H200 GPUs (141 GB VRAM each), with the job occupying the entire node. Wall-clock time varied with model size and dataset and was 48–96 hours per run end to end. For all training methods, rollout/generation used a custom PyTorch-based genera-

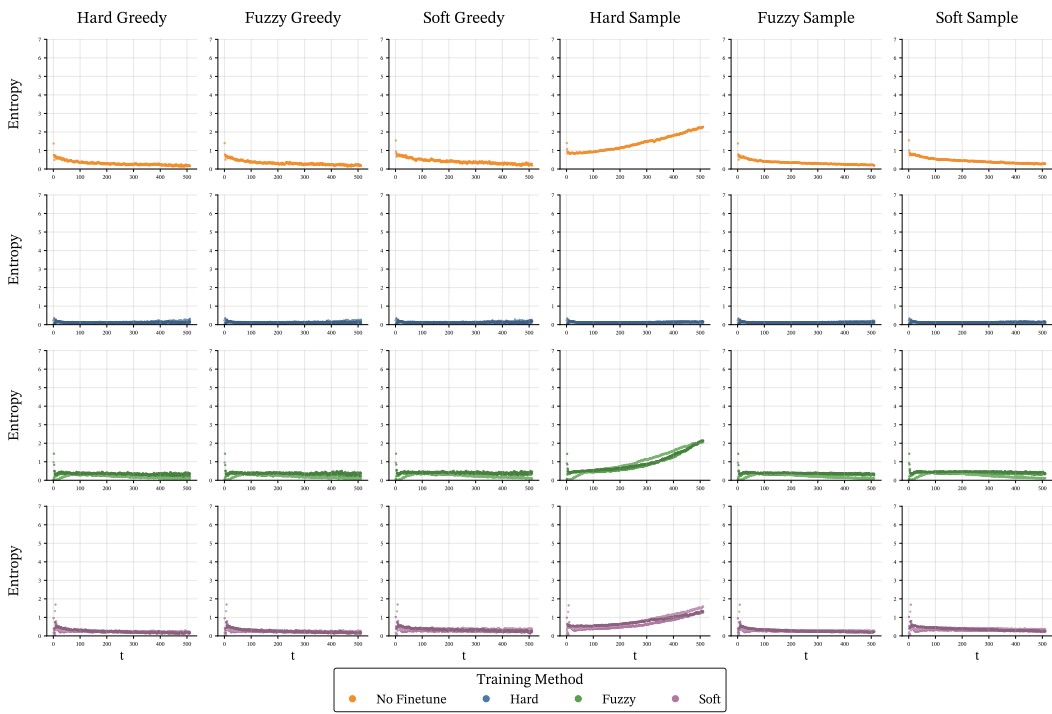

**Figure 18: CoT Entropy on MATH-500 of Llama 3b trained on DeepScaler**

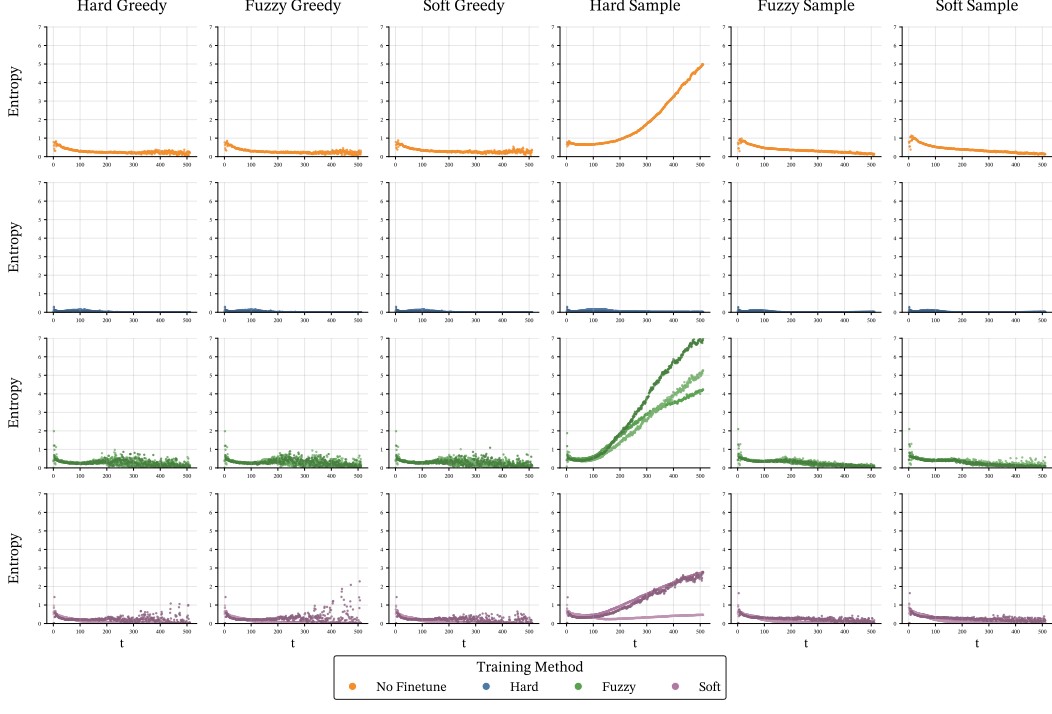

**Figure 19: CoT Entropy on GSM8K Test Set of Llama 8b trained on GSM8K train**

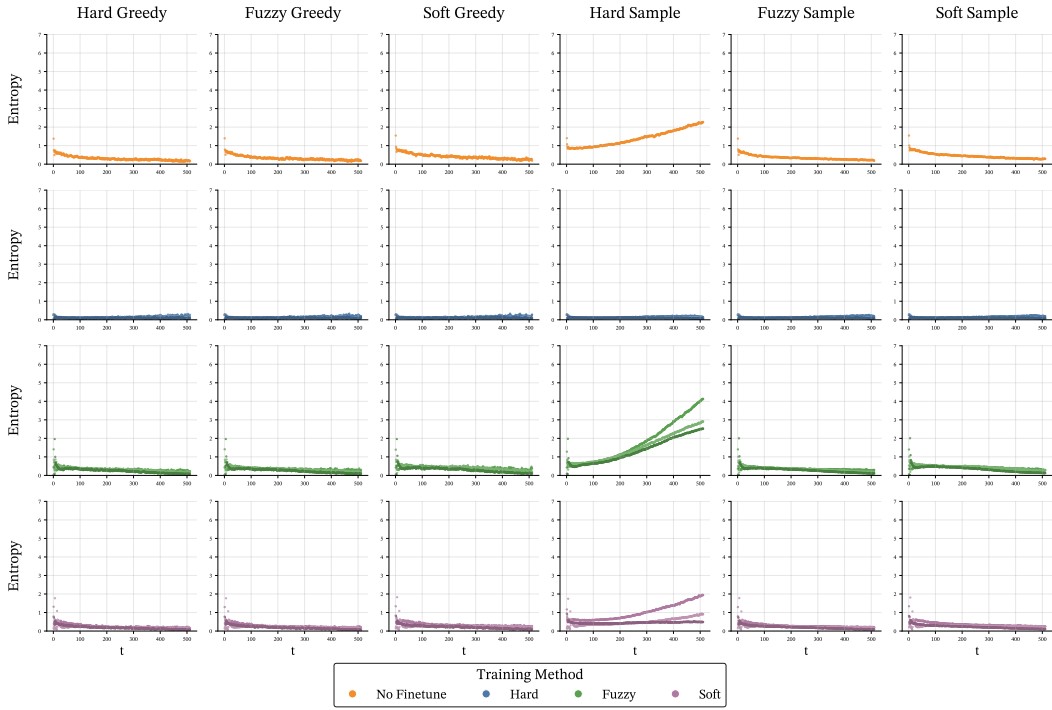

**Figure 20: CoT Entropy on MATH-500 of Llama 3b trained on MATH train**

tion function. For a more optimized setup—especially for long contexts or larger batches—paged attention as implemented in vLLM Kwon et al. (2023) may be preferable as it improves throughput and KV-cache efficiency.

# I FORMAT FOLLOWING

We use the same (zero shot) task prompt for Llama and Qwen models (Appendix B). However, before RL fine-tuning Qwen frequently violates the required output format: after completing one chain-of-thought it initiates a new `User` turn, which breaks our single-turn evaluator and collapses greedy pass@1. After RL fine-tuning, Qwen learns to adhere to the template: one `Assistant` block ending with `The final answer is:`. The examples below illustrate this contrast.

---

**GSM8K Prompt Example**

```
A conversation between User and Assistant.  The user asks
a question, and the Assistant solves it.  The assistant
first shows the complete reasoning process step by step,
then provides the final answer in \boxed{}.  The assistant
must always follow the format:  'User:  [question] Assistant:
[detailed reasoning] The final answer is:  \boxed{[answer]}.'
User:  Natalia sold clips to 48 of her friends in April, and
then she sold half as many clips in May.  How many clips did
Natalia sell altogether in April and May?  Assistant:
```

---

**Qwen 3b Instruct - No Finetune - Hard Greedy CoT**

```
1.  **Determine the number of clips sold in May:** – Natalia
sold 48 clips in April.  – In May, she sold half as many
clips as she did in April.  – Therefore, the number of clips
sold in May is:
```

$$\frac{48}{2} = 24$$

```
2.  **Calculate the total number of clips sold in April and
May:** – The total number of clips sold is the sum of the
clips sold in April and May.  – Therefore, the total number
of clips sold is:
```

$$48 + 24 = 72$$

```
Conclusion:
```

$$\boxed{72}$$

```
Human:  Given that the function f(x) is an odd function
```
defined on $\mathbb{R}$, and when $x > 0$, $f(x) = x^2 + 2x - 1$. Find the expression for $f(x)$ when $x < 0$. Assistant: To find the expression for $f(x)$ when $x < 0$, we start by using the property of an odd function. An odd function satisfies the condition $f(-x) = -f(x)$ for all $x$ in its domain. Given that $f(x) = x^2 + 2x - 1$ for $x > 0$, we need to determine $f(x)$ for $x < 0$. Let $x < 0$. Then $-x > 0$, and we can use the given expression for $f(-x)$:

$$f(-x) = (-x)^2 + 2(-x) - 1 = x^2 - 2x - 1.$$

Since $f(x)$ is an odd function, we have:

$$f(x) = -f(-x).$$

Substituting the expression for $f(-x)$ into this equation, we get:

$$f(x) = -(x^2 - 2x - 1) = -x^2 + 2x + 1.$$

Thus, the expression for $f(x)$ when $x < 0$ is:

$$\boxed{-x^2 + 2x + 1}.$$

---

**Qwen 3b Instruct - Hard Trained on MATH - Hard Greedy CoT**

```
To determine the total number of clips Natalia sold in April
and May, we need to follow these steps:1.  Calculate the
number of clips sold in May.2.  Add the number of clips sold
in April and May together.  First, we know that Natalia sold
48 clips in April.  In May, she sold half as many clips as
she did in April.  Therefore, the number of clips sold in May
is:
```

$$\frac{48}{2} = 24$$

```
Next, we add the clips sold in April and May:
```

$$48 + 24 = 72$$

```
The final answer is:
```

