# OpenReview forum: "Soft Tokens, Hard Truths"
_ICLR.cc/2026/Conference — ICLR 2026 Poster_

### Official Review · Reviewer_kauv · 2025-11-01

**Soundness:** 3
**Presentation:** 3
**Contribution:** 3
**Rating:** 6
**Confidence:** 3

**Summary:**

The paper proposes a simple, scalable way to train continuous Chain‑of‑Thought (CoT) with reinforcement learning by injecting Gaussian noise into probability‑mixture (“soft”/“fuzzy”) token embeddings during the CoT phase, making the thought trajectory stochastic and thus RL‑amenable. Using RLOO (leave‑one‑out REINFORCE) with Llama‑3 (3B/8B) and Qwen‑2.5 (3B) on math datasets (GSM8K, MATH, DeepScaleR), the authors report (i) pass@1 parity with discrete‑token CoT and (ii) consistent pass@32 gains, especially when training with soft/fuzzy CoT but performing hard (discrete) inference. They further show improved out‑of‑domain calibration (lower NLL on HellaSwag, ARC, MMLU) and an entropy analysis suggesting soft/fuzzy RL better preserves base‑model uncertainty than hard‑token RL. Figures 1–4 and Tables 1–2 (pp. 2, 7–9) summarize the core method and results.

**Strengths:**

* The paper is clearly written, well organized, and reproducibility details are ample (prompts, verifier, hyperparameters, etc). Figure 1 crisply contrasts hard/fuzzy/soft generation; Figures 2–4 and Tables 1–2 present results cleanly. Authors use newline a lot of time even in the abstract. Sometimes, it is useful, but it is abundant, it might be distracting. I respect authors' decision in general, but I wanted to add a comment on this.

* Conceptually, replacing discrete sampling with noisy mixtures to unlock RL for continuous CoT is neat and practical, requiring only noise at the embedding layer and storing $p_t$. Empirical gains are consistent in pass@32 and calibration, and the “train soft → infer hard” takeaway is actionable.

* Adding Gaussian noise to $pE$ (soft) or near‑one‑hot $p$ at tiny temperature (fuzzy) yields a tractable REINFORCE signal with minimal overhead.

**Weaknesses:**

* The paper says overhead is “minimal,” yet training uses G=32 samples per prompt and up to 512 CoT tokens, on 8×H100/H200 for 48–96 hours. A FLOPs comparison vs hard RL would strengthen the practicality claim.

* The related work is good but incomplete. I did not see any citation regarding adding noise to enable exploration, which has a decent literature. The authors can take a look at these papers: Jain et al., NEFTune. Yadav & Singh, SymNoise. Fortunato et al., Noisy Networks for Exploration. For latent thinking, the literature is growing very quickly. The authors can take a look at these papers as well: Gozeten et al., Continuous Chain of Thought Enables Parallel Exploration and Reasoning and Yue et al., Hybrid Latent Reasoning via Reinforcement Learning.

* For Llama‑3B on MATH, soft is ~4–5 points below hard. This means that pass@1 parity is not universal. The paper should nuance the claim and analyze when parity fails.

**Questions:**

* Do soft/fuzzy CoTs exhibit interpretable superpositions (e.g., linearly separable sub‑paths in $\tilde{h}_t^0)? Probing classifiers or SVCCA across diverse problems could show additional evidence about the mechanism. I am asking this question out of curiosity, and I wanted to know whether the authors tried probing before.

* Soft/fuzzy are behaving in parallel to each other. Do the authors find any cases where soft/fuzzy behaves slightly different?

---

> ### Author Response · Authors · 2025-11-22
> **Response to Reviewer kauv (1/2)**
>
> We thank you for your positive comments on the practicality of the method and pass@k performance.
>
> >The paper says overhead is “minimal,” yet training uses G=32 samples per prompt and up to 512 CoT tokens, on 8×H100/H200 for 48–96 hours. A FLOPs comparison vs hard RL would strengthen the practicality claim.
>
> We appreciate the opportunity to clarify this point.
>
> The reported wall-clock time is dominated by our custom, non-optimised PyTorch generation implementation.  For engineering reasons, we did not invest in optimising this code to match highly optimised inference engines such as vLLM, as our focus was on methodological questions. Therefore, the number of hours should not be taken as representative of deployment performance.
>
> The hard CoT RL runs use the same training duration (48–96 hours) as the soft/fuzzy ones, showing the soft/fuzzy overhead is minimal. The main additional computation in our soft/fuzzy setup is storing the probability vector \(p_t\) at each step and injecting noise at the first layer; both operations are cheap compared to a forward pass. As a result, the theoretical FLOPs of hard and soft/fuzzy training are extremely similar.
>
> >The related work is good but incomplete. I did not see any citation regarding adding noise to enable exploration, which has a decent literature. The authors can take a look at these papers: Jain et al., NEFTune. Yadav & Singh, SymNoise. Fortunato et al., Noisy Networks for Exploration. For latent thinking, the literature is growing very quickly. The authors can take a look at these papers as well: Gozeten et al., Continuous Chain of Thought Enables Parallel Exploration and Reasoning and Yue et al., Hybrid Latent Reasoning via Reinforcement Learning.
>
> We thank you for highlighting the very relevant literature on adding noise to enable exploration and concurrent works in latent reasoning.
>
> ''Continuous Chain of Thought Enables Parallel Exploration and Reasoning'' contains many results, but scalability is not established: the authors work with a 1-head GPT2 model on synthetic tasks, and in the main series of experiments the CoT is a binary string of length 4, resulting in just 16 possible CoTs. Presumably, any noise model that gives positive probability to the 16 possible CoTs would be able to succeed. Still, the Dirichlet noise model in their work is one we considered early on in our project, before settling on the simpler Gaussian noise on input embeddings.
>
> ''Hybrid Latent Reasoning via Reinforcement Learning'' introduces a curriculum that mixes continuous and discrete tokens via a convex combination, starting at mostly discrete and moving to mostly continuous. We believe this method would not work directly on full continuous tokens: then, their model becomes similar to Coconut and would face the same scalability issues (related to propagating the gradient backwards in time through the whole CoT as we used to do with backprop through time for recurrent networks). Our model, on the other hand, is simpler, and can work directly with full continuous tokens with minimal effort and no curriculum.
>
> The other references (NEFTune, SymNoise, and Noisy Networks) provide context for the beneficial effect of noise in various situations. NEFTune and SymNoise consider the SFT (non-RL, no CoT) finetuning of LLMs, and show that predicting the target sequence from a noised version of the input sequence leads to gains; this is somewhat similar to input perturbation as used for a long time (e.g. in the spirit of Denoising Autoencoders [2]). Noisy Networks indeed directly demonstrate the added benefits of noise on the model for providing diversity and exploration for RL.
>
> [2] Vincent, Pascal, et al. “Extracting and composing robust features with denoising autoencoders.” International Conference on Machine Learning (2008).
>
> >For Llama‑3B on MATH, soft is ~4–5 points below hard. This means that pass@1 parity is not universal. The paper should nuance the claim and analyze when parity fails.
>
> We agree and have updated the manuscript to reflect that pass@1 parity is not universal: we soften the parity claim and note the Llama-3B/MATH exception in the Figure 1 caption, results text, and conclusion (L69, L368-9, L481).
>
> >Do soft/fuzzy CoTs exhibit interpretable superpositions (e.g., linearly separable sub‑paths in $\tilde{h}_t^0)? Probing classifiers or SVCCA across diverse problems could show additional evidence about the mechanism. I am asking this question out of curiosity, and I wanted to know whether the authors tried probing before.
>
> We did not perform probing analyses in this work, but we agree that this would be a valuable way to study whether soft/fuzzy CoTs encode interpretable superpositions.

---

> ### Author Response · Authors · 2025-11-22
> **Response to Reviewer kauv (2/2)**
>
> >Soft/fuzzy are behaving in parallel to each other. Do the authors find any cases where soft/fuzzy behaves slightly different?
>
> We observe that, under hard sampling, the entropy profiles for fuzzy- and soft-trained models both resemble the base model, while entropy collapses for the hard-trained model (Figures 4, 17–19). Comparing fuzzy vs soft across different training datasets, we consistently see slightly higher entropy for the fuzzy-trained model, particularly towards the end of generation.
>
> We thank the reviewer again for their valuable comments and their time and hope that our responses address all issues raised. We would appreciate if you would consider raising your score.

---

### Official Review · Reviewer_PsKn · 2025-11-01

**Soundness:** 2
**Presentation:** 3
**Contribution:** 3
**Rating:** 6
**Confidence:** 4

**Summary:**

This paper introduces a reinforcement learning approach to train CoT with "soft" tokens—probabilistic mixtures of token embeddings with added noise for exploration without requiring distillation from reference answers. The study validates that models trained this way match standard discrete CoT models on pass@1 accuracy and consistently outperform at pass@32 (diversity), while also providing insights into entropy profiles and robustness on both in-domain mathematical reasoning tasks and out-of-domain tasks.

**Strengths:**

1. The soft CoT method proposed in the paper modestly improves pass@1 accuracy and significantly boosts pass@32 accuracy across multiple benchmarks. Meanwhile, both the soft and fuzzy approaches maintain pass@k curves that are close to the no-finetuning baseline when the value of k is large, preserving the model's foundational capabilities.

2. This paper propose a practical, memory-efficient method to train continuous CoT for hundreds of reasoning steps.The RL-based training approach eliminates the dependency on reference discrete tokens, thereby removing the constraints on CoT length that plagued previous work.

3. While achieving superior performance on the original datasets, soft and fuzzy methods have minimal impact on the base model's capabilities in domains outside mathematical reasoning, as evidenced by lower negative log-likelihood (NLL) on out-of-distribution tasks.

4. This paper conducts a detailed analysis of entropy, figure 4 demonstrates a notable distinction between entropy behavior under hard CoT and soft CoT RL, connecting lower entropy overconfidence to potential gains in reasoning diversity and generalization.

**Weaknesses:**

1. The performance improvement of soft CoT under pass@1 is not that significant. In some datasets and benchmarks, for example, when using Llama-3B-Instruct with DeepScalar on OlympiadBench—hard CoT seems to perform better. In this case, the pass@32 score of soft and fuzzy is higher, but it remains questionable whether this trade-off is worthwhile as both hard, soft, and fuzzy appear to have lower pass@32 scores than the base model.

2. The paper does not analyse group size, which is a crucial hyper parameter of RL training, which affects model entropy and pass@k performance. Soft CoT can be seen as considering multiple possibilities simultaneously during the CoT process, imposing less restriction on entropy compared to hard CoT, thus achieving better pass@k performance. However, could increasing the rollout group size of hard CoT also improve its pass@k?

3. Wang et al. (2022) (Self-Consistency for CoT) propose sampling multiple reasoning paths to improve  diversity. As pass@32 is a core claim, not including self-consistency as a baseline or in related work weakens the empirical case for the diversity benefit of soft CoT. While the paper discusses diversity and entropy, it's not clear whether the improvement is due to the token mixture, RL formulation, or just more stochasticity.

4. The main innovation is the practical integration of noise for RL compatibility,  rather than a fundamentally new architecture. However, the paper lacks direct  experimental comparison with distillation-based soft CoT methods to demonstrate  the advantages of the RL approach.

**Questions:**

1. Numerous related works have achieved considerable improvements in the field of RL + Math. Can fuzzy or soft CoT— which are orthogonal to RL methods— be applied to some of the current SOTA methods? Is the observed gain unique to continuous CoT RL, or mainly due to more diverse entropy? Would entropy  regularized hard CoT RL do as well?
2. Why were Llama-8b-Instruct and Qwen-3b-Instruct only trained on the GSM8K or Math datasets, instead of using all three training datasets (as was done for Llama-3b-Instruct)? Is the training method of soft CoT sensitive to the choice of dataset?
3. Have the authors explored generalization to non-math reasoning domains (e.g., code generation, commonsense multi-hop QA or other verifiable questions), do the claimed gains vanish as the underlying structure of reasoning becomes less sequential?

---

> ### Author Response · Authors · 2025-11-22
> **Response to Reviewer PsKn (1/2)**
>
> We thank you for highlighting the practicality of the method, pass@k performance, and additional analyses.
>
> >The paper does not analyse group size, which is a crucial hyper parameter of RL training, which affects model entropy and pass@k performance. Soft CoT can be seen as considering multiple possibilities simultaneously during the CoT process, imposing less restriction on entropy compared to hard CoT, thus achieving better pass@k performance. However, could increasing the rollout group size of hard CoT also improve its pass@k?
>
> We agree that rollout group size is an important hyperparameter. In this work, we fixed the group size across methods to isolate the effect of soft/fuzzy versus hard CoT under a comparable compute budget. Exploring how varying group size interacts with entropy and pass@k for these methods is an interesting direction for future work.
>
> >Wang et al. (2022) (Self-Consistency for CoT) propose sampling multiple reasoning paths to improve diversity. As pass@32 is a core claim, not including self-consistency as a baseline or in related work weakens the empirical case for the diversity benefit of soft CoT.
>
> We thank the reviewer for this suggestion and have added Wang et al. (2022) to the related work section to better contextualise our contributions. We do believe that measuring pass@k for various k, as we did, reflects on diversity. We agree that self-consistency is closely connected to our focus on pass@k and that a direct comparison would be informative. While we leave a full self-consistency evaluation for future work, our current results are at least consistent with an improvement in the diversity of useful reasoning paths rather than just increased randomness: our method largely maintains pass@1 parity with the hard CoT baseline, suggesting that probability mass on the most likely correct path is not degraded, while at the same time achieving higher pass@k. This combination of stable pass@1 (modal accuracy) and improved pass@k (coverage of additional plausible solutions) should, in principle, be favourable for self-consistency-style decoding, and we see exploring this connection empirically as a natural next step.
>
> >While the paper discusses diversity and entropy, it's not clear whether the improvement is due to the token mixture, RL formulation, or just more stochasticity.
>
> Regarding the source of the diversity gains, we can partially disentangle the factors using our existing experiments. First, we can confidently rule out the influence of stochasticity alone: this would correspond to step 0 in our training curves (before any RL training), which do not reach top performance.
>
> Second, the gains do not seem to come from the RL formulation alone, since soft/fuzzy RL outperforms hard CoT RL in terms of diversity. Finally, the fuzzy formulation uses a much more concentrated token mixture (due to very low temperature) yet exhibits similar improvements to the softer mixtures, suggesting that token mixture per se is not the *only* driver. Given these observations, the remaining difference between methods is how noise is applied: our results are therefore most consistent with the gains arising from *how* stochasticity is introduced (embedding-level noise during RL training), rather than from simply increasing the amount of randomness.
>
> >The main innovation is the practical integration of noise for RL compatibility, rather than a fundamentally new architecture. However, the paper lacks direct experimental comparison with distillation-based soft CoT methods to demonstrate the advantages of the RL approach.
>
> Our primary goal is to develop an RL-based approach that does not require reference CoTs, so our main comparison is with hard CoT RL. Distillation-based soft CoT methods rely on supervised trajectories, whereas our setting explicitly assumes their absence. Thus, a like-for-like comparison would be very challenging (if not unfair) and we consider this beyond the scope of the current work.
>
> >Numerous related works have achieved considerable improvements in the field of RL + Math. Can fuzzy or soft CoT— which are orthogonal to RL methods— be applied to some of the current SOTA methods?
>
> You are right to point out that our model is orthogonal to many of the RL refinements recently introduced, and could potentially be combined. This could be tackled in future work.
>
> For all models (base model, hard RL, fuzzy RL, soft RL), we perform hard, soft and fuzzy inference, as reported in Appendix E. Contrary to prior work, we see no benefit to soft and fuzzy inference over hard inference. In this work, we only observe benefit from fuzzy/soft RL training.

---

> ### Author Response · Authors · 2025-11-22
> **Response to Reviewer PsKn (2/2)**
>
> >Is the observed gain unique to continuous CoT RL, or mainly due to more diverse entropy? Would entropy regularized hard CoT RL do as well?
>
> We did not implement entropy-regularised hard CoT RL in this work, so we cannot make a direct empirical comparison, but we view combining our approach with stronger entropy-regularised RL baselines as an interesting direction for future work.
>
> >Why were Llama-8b-Instruct and Qwen-3b-Instruct only trained on the GSM8K or Math datasets, instead of using all three training datasets (as was done for Llama-3b-Instruct)? Is the training method of soft CoT sensitive to the choice of dataset?
>
> We were constrained by compute limitations. We chose to run LLama-8b-Instruct on GSM8K because the number of CoT tokens required is lower so it used less compute. We chose to run Qwen-3b-Instruct only on MATH because we suspect that Qwen was trained on GSM8K already. All method/dataset combinations that we ran are reported in the paper and we observe fairly consistent results across training and test sets.
>
> >Have the authors explored generalization to non-math reasoning domains (e.g., code generation, commonsense multi-hop QA or other verifiable questions), do the claimed gains vanish as the underlying structure of reasoning becomes less sequential?
>
> Due to compute limitations, we did not explore generalization to non-math reasoning domains.
>
> We thank the reviewer again for their valuable comments and their time and hope that our responses address all issues raised. We would appreciate if you would consider raising your score.

---

### Official Review · Reviewer_gSxV · 2025-11-01

**Soundness:** 2
**Presentation:** 4
**Contribution:** 3
**Rating:** 6
**Confidence:** 3

**Summary:**

The paper presents an algorithm for LM post-training that leverages the continuous space of hidden states to encourage exploration. This approach achieves performance comparable to or better than standard token-based methods in pass@1, and attains the best results on pass@k for larger values of k (compared to the hard baseline), suggesting greater diversity in the model’s outputs.

**Strengths:**

- The method is simple and can be easily implemented for broad adaptation.
- The analysis is thorough, covering out-of-distribution (OOD) tasks and multiple models.
- The presentation and illustrations are clear and effectively aid understanding of the method.

**Weaknesses:**

- Nearly all reported results overlap with the hard baseline (at least for pass@1). It remains unclear why this method is preferable—for instance, one could dynamically switch between a tuned model and a “No train” model depending on the required inference budget (k).
- The authors avoid a direct comparison with [1]. Although they cite it in the related work, it seems relevant to consider whether training with that method’s type of noise—and then inferring it as soft or hard tokens—would yield similar benefits. Notably, [1] was published in August 2025, making the works concurrent; thus, a detailed comparison may not be expected.
- In Figure 3, I recommend adding pass@64. It would be valuable to see whether the “No Finetune” baseline eventually surpasses the Soft and Fuzzy methods at higher k.

[1] LLMs are Single-threaded Reasoners: Demystifying the Working Mechanism of Soft Thinking. Junhong Wu et al.

**Questions:**

See weaknesses

---

> ### Author Response · Authors · 2025-11-22
> **Response to Reviewer gSxV**
>
> We thank you for your appreciation of the simplicity of the method and the quality of the analysis and presentation.
>
> >Nearly all reported results overlap with the hard baseline (at least for pass@1). It remains unclear why this method is preferable—for instance, one could dynamically switch between a tuned model and a “No train” model depending on the required inference budget (k).
>
> This is a fair point. However, in many deployment settings it is substantially simpler to maintain and serve a single model rather than dynamically switching between differently tuned variants at inference time.
>
> >The authors avoid a direct comparison with [1]. Although they cite it in the related work, it seems relevant to consider whether training with that method’s type of noise—and then inferring it as soft or hard tokens—would yield similar benefits. Notably, [1] was published in August 2025, making the works concurrent; thus, a detailed comparison may not be expected.
>
> We appreciate this suggestion. As described in the global response, we have added a new ablation using Gumbel noise on the logits, to better connect our results to [1]. However, we find that, similarly to using Gaussian noise, there is no evidence of RL learning when the Gumbel noise is placed on the output logits. Like with Gaussian noise on the logits (Appendix F.2), we hypothesise that this is due to the dimensionality of the noise being higher when placed on logits than on input embeddings, resulting in a lower signal to noise ratio. This new ablation may be found in the updated manuscript in Appendix F.3.
>
> >In Figure 3, I recommend adding pass@64. It would be valuable to see whether the “No Finetune” baseline eventually surpasses the Soft and Fuzzy methods at higher k.
>
> We agree that extending the pass@k curves to higher k is informative. During the rebuttal period, we extended the analysis for Llama-3B-Instruct trained on GSM8K to pass@64 and we observe soft and fuzzy training methods achieve similar pass@64 to the no-finetune base model, whereas the hard training method continues to under-perform. These results are now included in Appendix G.3.
>
> We thank the reviewer again for their valuable comments and their time and hope that our responses address all issues raised. We would appreciate if you would consider raising your score.

---

### Official Review · Reviewer_vwfP · 2025-11-05

**Soundness:** 3
**Presentation:** 3
**Contribution:** 2
**Rating:** 4
**Confidence:** 3

**Summary:**

This paper focuses on designing a scalable training approach for continuous Chain-of-Thoughs (CoT). Towards this, the paper proposes a reinforcement learning method that feeds a "soft token" produced by an LLM as the input to the LLM at the next step after adding random Gaussian noise to the soft token. The paper also considers a variant of this approach which feeds a noisy "fuzzy token" as opposed to the noisy "soft token" as the input. The paper claims that they propose the first continuous CoT training approach that does not require ground-truth discrete CoT. The proposed method achieves pass@1 accuracy which is on-par that of discrete CoT while realizing higher pass@K performance. Furthermore, the proposed training approach ensures that the base model's out of domain generalization is unaffected. The experiments are conducted with the Llama 3 and Qwen 3 family models on multiple datasets, including GSM8K, MATH-500, and OlympiadBench.

**Strengths:**

- The paper proposes a scalable training method for continuous CoT.
- The proposed method achieves better pass@K than discrete CoT.
- The proposed method preserves model performance on out of domain tasks.

**Weaknesses:**

- The paper does not provide an adequate discussion of prior works that propose reinforcement learning approaches for continuous CoT. E.g., see https://arxiv.org/pdf/2505.18454 and https://arxiv.org/abs/2505.23648. Could authors provide a discussion on why their approach adds significant novelty on top of such existing works?
- The main contribution of the paper appears to be limited to adding the noise to soft/fuzzy tokens before feeding it as an input. However, the paper does not provide a detailed treatment of this approach. Why is Gaussian choice the best approach? Even within the Gaussian noise, how noise parameters affect the overall performance.
- The paper shows that soft/fuzzy training followed by hard sampling (discrete CoT) performs the best. However, the paper does provide any meaningful analysis/discussion on why such a discrepancy between the training and test time leads to the best performance.
- It appears that for pass@1, continuous CoT does not outperform discrete CoT. But does continuous CoT provide inference efficiency benefits such as smaller thought lengths, especially when continuous CoT is employed at the test time?

**Questions:**

See the weaknesses section.

---

> ### Author Response · Authors · 2025-11-22
> **Response to Reviewer vwfP (1/2)**
>
> We thank you for highlighting the scalability of our method, the higher pass@k compared to discrete tokens, and the performance preservation on out-of-domain tasks.
>
> > The paper does not provide an adequate discussion of prior works that propose reinforcement learning approaches for continuous CoT. E.g., see https://arxiv.org/pdf/2505.18454 and https://arxiv.org/abs/2505.23648.
>
> We thank you for these references to very relevant concurrent works and have included them in the updated manuscript.
>
> > Could authors provide a discussion on why their approach adds significant novelty on top of such existing works?
>
> We believe these works do not demonstrate a scalable method for learning fully continuous CoTs.
>
> ''Continuous Chain of Thought Enables Parallel Exploration and Reasoning'' contains many results, but scalability is not established: the authors work with a 1-head GPT2 model on synthetic tasks, and in the main series of experiments the CoT is a binary string of length 4, resulting in just 16 possible CoTs. Presumably, any noise model that gives positive probability to the 16 possible CoTs would be able to succeed. Still, the Dirichlet noise model in their work is one we considered early on in our project, before settling on the simpler Gaussian noise on input embeddings. Our ablations in Appendix F.2 show that adding Gaussian noise to high-dimensional logits hinders RL training. Instead, we found that adding noise to the input embeddings, rather than the output logits, is necessary to successfully train CoTs with hundreds of tokens.
>
> ''Hybrid Latent Reasoning via Reinforcement Learning'' introduces a curriculum that mixes continuous and discrete tokens via a convex combination, starting at mostly discrete and moving to mostly continuous. We believe this method would not work directly on full continuous tokens: then, their model becomes similar to Coconut and would face the same scalability issues (related to propagating the gradient backwards in time through the whole CoT as we used to do with backprop through time for recurrent networks). Our model, on the other hand, is simpler, and can work directly with full continuous tokens with minimal effort and no curriculum.
>
> > The main contribution of the paper appears to be limited to adding the noise to soft/fuzzy tokens before feeding it as an input. However, the paper does not provide a detailed treatment of this approach...Even within the Gaussian noise, how noise parameters affect the overall performance.
>
> The main technical novelty is the combination of (i) injecting noise at the input embedding level of soft/fuzzy tokens, and (ii) modifying the REINFORCE objective and gradient flow accordingly. The location of the noise is crucial: by injecting it into the embeddings, we stop gradients at a different point than in traditional REINFORCE on discrete tokens.
>
> We do provide additional detail on the influence of noise parameters in Appendix F. In Appendix F.1, we ablate the noise scale and find that performance is robust for scale factors below 1; when the scale is substantially larger than the RMS embedding norm, training collapses. In Appendix F.2, we compare different noise placements (e.g., logits vs embeddings) and find that placing noise on the input embeddings is the only setting that consistently leads to effective learning. Finally, as discussed in the global response, we have included an additional ablation where we use Gumbel noise on the logits instead of Gaussian noise for the noise placement ablations. With both types of noise, we observe no evidence of learning when the noise placement is on the logits instead of input embeddings. This new ablation may be found in the updated manuscript in Appendix F.3.
>
> > Why is Gaussian choice the best approach?
>
> As discussed in the global response, our choice is motivated by the noise placement. Since we add noise to the input embeddings, the model first multiplies these noisy embeddings by a weight matrix, yielding a large weighted sum of noise components. This aggregated sum often tends toward a Gaussian distribution even if the original noise is non-Gaussian, making Gaussian noise a natural and simple choice.

---

> ### Author Response · Authors · 2025-11-22
> **Response to Reviewer vwfP (2/2)**
>
> > The paper shows that soft/fuzzy training followed by hard sampling (discrete CoT) performs the best. However, the paper does provide any meaningful analysis/discussion on why such a discrepancy between the training and test time leads to the best performance.
>
> We hypothesize that soft/fuzzy training forces the model to learn a reasoning manifold that is robust to local perturbations. At test time, hard sampling allows the model to traverse the highest-probability path through this optimized manifold, yielding the best performance. The noise during training ensures the model does not collapse to a narrow, brittle solution, which explains the diversity gains. Presumably, building on the "superposition of paths" supported by several related works, continuous tokens allow the model to keep a superposition of hypotheses during training, while hard-token inference lets it concentrate on the single best path at inference time.
>
> > It appears that for pass@1, continuous CoT does not outperform discrete CoT. But does continuous CoT provide inference efficiency benefits such as smaller thought lengths, especially when continuous CoT is employed at the test time?
>
> We examined the lengths of the CoTs under all methods, during both training and inference, across all models and did not observe consistent differences between settings.
>
> We thank the reviewer again for their valuable comments and their time and hope that our responses address all issues raised. We would appreciate if you would consider raising your score.

---

### Author Response · Authors · 2025-11-22
**Global Response**

# Global Response
We thank reviewers for their detailed and constructive feedback. All reviewers highlight the practicality, scalability, or simplicity of our method, as well as the improved pass@k performance and the analysis on out-of-domain tasks. Some also comment positively on the clarity of the presentation and the entropy-based analysis of different training schemes.

We group common points below and respond to reviewer-specific comments in the corresponding sections. We have updated the manuscript based on these comments, with changes shown in blue.

## Related Work (vwfP, kauv)
We thank both reviewers vwfP and kauv for highlighting very relevant concurrent works in latent reasoning. Further, we also appreciate reviewer kauv's suggestion of expanding the related work to include papers on adding noise to enable exploration. We describe the main differences with concurrent work in the reviewer-specific answers below. We have updated the related work section in the manuscript accordingly.

## Gaussian Noise, Noise Location, & New Ablation (vwfP, gSxV)
Reviewers vwfP and gSxV raised questions about our choice of Gaussian noise. Reviewer vwfP asks, "Why is Gaussian choice the best approach?", and reviewer gSxV comments, "it seems relevant to consider whether training with that method’s [1] type of noise—and then inferring it".

The placement of the noise in the model (input embeddings, final hidden layer outputs, or logits) is crucial for this choice. In our setup, noise is added to the input embeddings. Because these noisy embeddings are immediately multiplied by weight matrices, the model effectively observes a weighted sum of many random variables. This sum often tends toward a Gaussian distribution even when the individual noises are coming from a different distribution, which makes Gaussian noise a natural and simple choice in this setting.

In concurrent work [1], the authors find that soft inference alone (on hard-trained models) does not outperform hard inference; however, upon adding Gumbel noise to the model logits (on hard-trained models), they observe an improvement compared to temperature sampling. At the beginning of this project, we experimented with soft inference on hard-trained models, adding different noises (Gumbel and Gaussian) to the logits. However, in our implementation we did not observe improvements over hard tokens, so we did not pursue this direction further.

When we moved beyond inference and began training on soft CoT, our first approach was to add noise to the logits and train with REINFORCE. This ablation is detailed in Appendix F.2: with Gaussian noise on the output logits, we observe no evidence of effective learning. We hypothesised that this was due to the dimensionality of the noise: on the logits, the noise vector has vocabulary size, whereas on the input embeddings it has the hidden dimension. This likely leads to a worse signal-to-noise ratio. Thus, we also experimented with a top-$k$ variation on logit noise, where we saw some evidence of learning for low $k$ values.

To better connect to [1], we have added an additional ablation where we replace the Gaussian noise on the logits in the noise placement ablations with Gumbel noise, for RL training and inference. Similarly to the ablation with Gaussian noise on the logits, we do not observe effective learning.

We thank both reviewers for their insightful comments and we have added the new ablation to the manuscript in Appendix F.3.

[1] Wu, Junhong, et al. "LLMs are Single-threaded Reasoners: Demystifying the Working Mechanism of Soft Thinking." arXiv preprint arXiv:2508.03440 (2025).

---

### Author Response · Authors · 2025-12-03
**Final Summary to AC**

Dear Area Chair,

We thank the reviewers for their constructive feedback and the time invested in reviewing our work. We are encouraged that the reviewers found our approach to be novel, scalable, and practical.

We briefly summarize the consensus on the strengths of the paper and the key updates made during the rebuttal:

1. **Scalability and Practicality (Reviewers vwfP, gSxV, PsKn, kauv)**
   Reviewers unanimously highlighted that our method addresses a critical gap in the literature: scalability.
   - Reviewer vwfP noted we propose the "first continuous CoT training approach that does not require ground-truth discrete CoT."
   - Reviewer PsKn emphasized that our method is "memory-efficient" and "eliminates the dependency on reference discrete tokens."
   - Reviewer kauv described the method as "neat and practical," offering a "minimal overhead" solution.

2. **Empirical Gains: Diversity & Robustness (Reviewers PsKn, gSxV, vwfP, kauv)**
   The reviews emphasise that soft RL training improves reasoning diversity and robustness.

   - **Superior Pass@k Performance:** Reviewers highlighted that our approach "significantly boosts pass@32 accuracy" (Reviewer PsKn) and "attains the best results on pass@k for larger values of k" (Reviewer gSxV).
   - **OOD Generalization:** Reviewers vwfP and PsKn also highlighted that our method preserves the base model's "out of domain generalization", maintaining performance on non-math tasks (HellaSwag, ARC, MMLU).

3. **Rebuttal Updates**
   The primary concern raised was the context of concurrent work. In our updated manuscript, we have addressed this directly:

   - **Expanded Related Work (Reviewers vwfP, kauv):** We have significantly updated the Related Work section to include concurrent latent reasoning papers and literature on noise for exploration. We explicitly clarified that our work adds significant novelty over these concurrent latent reasoning approaches because we are the only ones to demonstrate scalability to hundreds of fully continuous tokens, whereas prior methods are limited to short sequences or require complex mixing curricula.
   - **New Ablation on Noise Placement (Reviewers vwfP, gSxV):** To address comparisons with concurrent work [1], we added an ablation using Gumbel noise on output logits (Appendix F.3). The results confirm that injecting noise at the input embedding level is essential for effective learning in our setting.
   - **Extended Analysis (Reviewer gSxV):** We added Pass@64 results (Appendix G.3), demonstrating that our performance advantage over the hard baseline holds even at higher sample budgets.

We believe this work offers a simple and scalable path forward for latent reasoning in LLMs.

---

### Meta-Review · Area_Chair_VU3g · 2026-01-06

**Summary:**

The paper makes a solid contribution by demonstrating the first scalable approach to continuous CoT training via RL, with hundreds of reasoning tokens on 3B-8B models using standard hardware and no curriculum. The consistent diversity gains (pass@k) and preserved OOD capabilities address real limitations of hard-token RL.

Reviewers unanimously praised the method's scalability, practicality, and simplicity. Initial concerns focused on positioning relative to concurrent work, justification for the noise strategy, and missing higher-k analysis.
The rebuttal strengthened the paper substantially through targeted ablations and expanded contextualization. Outstanding concerns are reasonable limitations for future work rather than fundamental flaws.

**Reviewer Concerns:**

*Addressed by rebuttal*:
- **Concurrent work and novelty (vwfP, kauv)**: Authors expanded related work and articulated the key differentiator: scalability to hundreds of fully continuous tokens without mixing curricula or backprop-through-time constraints.
- **Noise choice and placement (vwfP, gSxV)**: Authors provided theoretical justification (CLT in embedding space) and added a new ablation (Appendix F.3) showing Gumbel noise on logits fails to learn, validating embedding-level placement as essential.
- **Higher-k analysis (gSxV)**: Pass@64 results added (Appendix G.3), confirming soft/fuzzy advantages persist.
- **Pass@1 parity claims (kauv)**: Authors nuanced claims to acknowledge exceptions (e.g., Llama-3B on MATH).
- **Overhead clarification (kauv)**: Authors explained wall-clock time reflects unoptimized code; theoretical FLOPs are similar to hard RL.

*Outstanding (limitations, not fatal flaws)*:
- **Self-consistency baseline (PsKn)**: Citation added but no direct experimental comparison.
- **Rollout group size ablation (PsKn)**: Fixed across methods; interaction with entropy unexplored.
- **Non-math generalization (PsKn)**: Benefits on code or QA domains untested.
- **FLOPs quantification (kauv)**: Argument is plausible but not empirically measured.

**Reviewer Scores:**

- **Reviewer vwfP (initial: 4)**: likely increase to **6**. Primary concerns (novelty vs. concurrent work, noise rationale) were directly addressed with citations and new ablations. Some explanations remain qualitative, limiting increase beyond 5.
- **Reviewer gSxV (initial: 6)**: likely remain at **6** or slight increase. Both requested items (pass@64, Gumbel ablation) were provided.
- **Reviewer PsKn (initial: 6)**: likely remain at **6**. Core contribution appreciated; remaining requests (self-consistency, group size) acknowledged as future work.
- **Reviewer kauv (initial: 6)**: likely remain at **6**. Related work and pass@1 nuancing addressed; overhead argument is plausible though unquantified.

---

### Decision · Program_Chairs · 2026-01-26

Accept (Poster)